# Data-driven fine-grained region discovery in the mouse brain with transformers

Alex J. Lee[1,2], Alma Dubuc [1,2], Michael Kunst [3], Shenqin Yao [3], Nicholas Lusk[3], Lydia Ng[3], Hongkui Zeng [3], Bosiljka Tasic [3] & Reza Abbasi-Asl [1,2,4] ✉

Spatial transcriptomics offers unique opportunities to define the spatial organization of tissues and organs, such as the mouse brain. We address a key bottleneck in the analysis of organ-scale spatial transcriptomic data by establishing a workflow for self-supervised spatial domain detection that is scalable to multimillion-cell datasets. This workflow uses a self-supervised framework for learning latent representations of tissue spatial domains or niches. We use an encoder-decoder architecture, which we named Cell-Transformer, to hierarchically learn higher-order tissue features from lower-level cellular and molecular statistical patterns. Coupling our representation learning workflow with minibatched GPU-accelerated clustering algorithms allows us to scale to multi-million cell MERFISH datasets where other methods cannot. CellTransformer is effective at integrating cells across tissue sections, identifying domains highly similar to ones in existing ontologies such as Allen Mouse Brain Common Coordinate Framework (CCF) while allowing discovery of hundreds of uncataloged areas with minimal loss of domain spatial coherence. CellTransformer domains recapitulate previous neuroanatomical studies of areas in the subiculum and superior colliculus and characterize putatively uncataloged subregions in subcortical areas, which currently lack subregion annotation. CellTransformer is also capable of domain discovery in whole-brain Slide-seqV2 datasets. Our workflows enable complex multi-animal analyses, achieving nearly perfect consistency of up to 100 spatial domains in a dataset of four individual mice with nine million cells across more than 200 tissue sections. CellTransformer advances the state of the art for spatial transcriptomics by providing a performant solution for the detection of fine-grained tissue domains from spatial transcriptomics data.

Hierarchical spatial organization is ubiquitous in tissue and organ biology. Systematic, high-dimensional phenotypic measurements of this organization, generated through experimental tools such as spatial transcriptomics, multiplex immunofluorescence, and electron microscopy, are also becoming increasingly available as large, open datasets. However, transforming this abundance of data into a useful representation can be difficult, even for fields with a wealth of prior knowledge, such as neuroanatomy.

[1]Department of Neurology, University of California, San Francisco, CA, USA. [2]UCSF Weill Institute for Neurosciences, San Francisco, CA, USA. [3]Allen Institute for Brain Science, Seattle, WA, USA. [4]Department of Bioengineering and Therapeutic Sciences, University of California, San Francisco, CA, USA. ✉e-mail: Reza.AbbasiAsl@ucsf.edu

Datasets such as the Allen Brain Cell Whole Mouse Brain (ABC-WMB) Atlas[1–3], a multi-million cell single-cell RNA sequencing (scRNA-seq) and spatial transcriptomics (MERFISH) atlas, provide unprecedented opportunities to investigate whether computational tools can help biologists understand spatial cellular and molecular organization. However, the size of these datasets presents computational challenges for existing methods. Existing methods for spatial niche or spatial domain detection often operate on the entire dataset at once, for example, a tissue-section-wide cell by gene matrix. This precludes scale-up to large multi-section datasets as most systems do not have the GPU memory required to load multiple sections of data or store intermediary representations such as pairwise distance matrices[4–6], particularly as datasets scale into the millions or tens of millions of cells. Some methods rely on Gaussian processes, which feature a costly cubic computational scaling in the number of observations[7]. Other more scalable methods are limited in capturing granular structure, integration across tissue sections, or require significant neuroanatomical prior knowledge to manually audit, cluster, and hyperparameter tune for domain discovery workflows[8–10].

Our method, CellTransformer, implements a robust representation learning and clustering workflow to discover spatial niches at scale by representing not tissue sections but subgraphs that represent individual cellular neighborhoods. We describe an innovative strategy to induce the encoder of an encoder-decoder transformer to aggregate useful information into a neighborhood representation token. This occurs by training the model to condition cell-type specific gene expression predictions using this neighborhood context token. The model thus learns to predict the expression of cell types in arbitrary cell neighborhoods. This representation allows for recovery of important anatomically plausible spatial domains while remaining computationally efficient.

We evaluate CellTransformer on using the ABC-WMB dataset (3.9 million cells collected with a 500 gene MERFISH panel)[1] demonstrating its effectiveness in producing completely data-driven spatial domains of the mouse brain by comparing the results to the Allen Mouse Brain Common Coordinate Framework version 3 (CCFv3)[11]. CCF is a consensus hand-drawn 3D reference space compiled from a large multimodal data corpus. Annotations feature labels at three levels of coarseness (from 25 regions at coarse-grain to 670 at fine-grain), which we use to show that CellTransformer excels at identifying spatial domains that are spatially coherent and biologically relevant. CellTransformer domains reproduce known regional architecture observed in targeted studies of the subiculum and in the superior colliculus superficial layers. Beyond the 670 regions currently annotated in ABC-WMB, we show our workflow produces meaningful data-driven domains in regions that currently lack subregion annotation. As examples, we focus on data-driven subdomains we define in superior colliculus and midbrain reticular nucleus.

We also demonstrate CellTransformer's strength in integrating domains across animals, leveraging a separate whole-brain dataset within ABC-WMB[12] comprising 6.5 million cells distributed across four animals and 239 sections and with a separate gene panel with 1129 genes. We find that CellTransformer produces consistent subregions across all 5 animals (1 coronal and 4 sagittal), suggesting a successful integration across animals with heterogeneous measurements. Notably, we also find that identified domains are highly consistent across animals. This work demonstrates that large-scale data-driven discovery of domains at CCF-like resolution can be based on spatial transcriptomics data. Finally, we show that our framework can perform domain detection in a different spatial transcriptomics modality, Slide-seqV2, using a whole-brain dataset of cellularly deconvoluted results[13].

## Results

### The CellTransformer architecture and domain detection workflow

CellTransformer is a graph transformer[14] neural network that is trained to learn latent representations of cell neighborhoods by conditioning single-cell gene expression predictions on neighborhood spatial context. We define a cellular neighborhood as any cells within a user-specified distance cutoff in microns away from a reference or center cell. As input, our model requires the gene expression profiles and cell type classifications for cells in a neighborhood and outputs a latent variable representation for that neighborhood. One of the principal operations in a transformer is the self-attention operation, which computes a feature update based on pairwise interactions between elements in a sequence, which are referred to as tokens (here, cells). Accordingly, one interpretation of our model is of learning an arbitrary and dynamic pairwise interaction graph among cells.

Restricting this graph to a small neighborhood subgraph of the whole-tissue-section graph has benefits for both computational resource usage and biological interpretability. We interpret the size of the neighborhood as a constraint on the physical distance at which statistical correlations between the observed cells and their gene expression profiles can be directly captured. Truncating neighborhoods using a fixed spatial threshold instead of choosing a fixed number of neighbors also allows the network to account for the varying density of cells in space. Accordingly, our framework incorporates a notion of both cytoarchitecture (relative density and proximity) and molecular variation (cell type and RNA-level variation) in the data.

We designed a self-supervised training scheme requiring only cell-type labels, which many large-scale studies make available via scRNA-seq atlas reference mapping[1,12]. We model cellular neighborhoods as sets of cell tokens that are within a box of fixed size centered around a reference cell and use them to predict the observed gene expression of the cell at the center of the neighborhood. We refer to this cell as the reference cell (indicated by "cell R" in Fig. 1a). Cell tokens are generated by composing cell-type and gene expression information (*Methods*). After encoding with a series of transformer layers (where cells are only allowed to attend to each other if they are in the same neighborhood), these tokens are then aggregated using a learned pooling operation to produce a single token representation of the entire tissue context. The model receives a new mask token representing the reference cell's type, which is used to predict its gene expression following the operation of several transformer decoder layers (Fig. 1b; note that the point cloud displayed is for illustration purposes only). Importantly, during this process, only the mask token and the neighborhood representation can attend to each other. This operation captures a hierarchical encoding and decoding process where low-level information (gene and cell type) is produced at the cell token level and aggregated into a high-level representation. This high-level representation is then used to conduct the reverse decoding process (prediction of gene expression from cell type and tissue context information). This construction resembles that of masked language models[14] and masked autoencoders[15,16], where masked predictions are generated based on a conditioning signal (position encodings). In our model, position-based conditioning is replaced by cell type conditioning, similarly to the NCEM[17] model. However, unlike masked prediction models such as NCEM[17], we aggregate information across tokens (cells) in a cellular neighborhood using a learned pooling, which strongly bottlenecks the information distributed across the tokens prior to masked cell prediction.

At test time, we extract this neighborhood representation for each cell and use *k*-means clustering to identify discrete spatial domains (Fig. 1c). We will use the term spatial domain to refer to the output of clustering on embeddings and cluster to refer to single-cell clusters transferred from the ABC-WMB single-cell taxonomy. We emphasize

that the input embedding matrix for *k*-means is constructed by concatenating all cells across the dataset across tissue sections. Since minibatching is used during training (unlike methods such as STAligner and GraphST), for generating embeddings, and during *k*-means (using cuml for GPU-acceleration), overall computational costs of our algorithm are limited in principle only by the memory required for storage of cellular neighborhoods rather than entire sections or datasets.

## Data-driven discovery of fine-grained spatial domains in the mouse brain using ABC-WMB

The ABC-WMB spatial transcriptomics dataset contains data from five mouse brains[1,12]. One animal was processed by the Allen Institute for Brain Science with a 500 gene MERFISH panel and 53 coronal sections (Yao et al.)[1] The remaining four other animals, generated in Zhang et al.[11]. were collected with a 1129 gene panel. Sections from two of these animals ("Zhuang 1", 147 sections; and "Zhuang 2", 66 sections) were sampled coronally. The other two animals in the dataset ("Zhuang 3", 23 sections; and "Zhuang 4", 3 sections) were sampled sagittally.

We first trained CellTransformer on the Allen 1 dataset. A visualization of the loss curve (negative log-likelihood) showed a clear

plateau (Supplementary Fig. 1). We then clustered these embeddings using *k*-means. We emphasize that to generate spatial domains across the brain, all *k*-means clustering in this paper was performed by concatenating cells in the dataset across tissue sections. All further references to visualizations of domains, including those only visualized for a subset of domains, were fit at a given number of domains across the entire dataset. We observed an isolated case where a small number of spatial clusters only in the striatum displayed a non-homogenous spatial patterning that strongly resembled a predominantly astrocytic, *Crym+* population described in Ollivier et al. (2024) and have similar gene expression patterns (particularly *Crym* expression) and spatial organization (See Supplementary Note 1 for a discussion on biological plausibility of these domains and the effect of smoothing, as well as an extensive series of visualizations of the domains at high resolution with and without smoothing). Despite this plausibility, we introduced an optional smoothing step (see *Methods*) to remove this population to concord with neuroanatomical convention, which typically analyzes spatially contiguous regions. We also visualized the first three principal components (~19.5% variance explained, Supplementary Fig. 2a–c), demonstrating the CellTransformer embeddings produce neuroanatomically reasonable spatial patterns. For example, PC1 and PC2

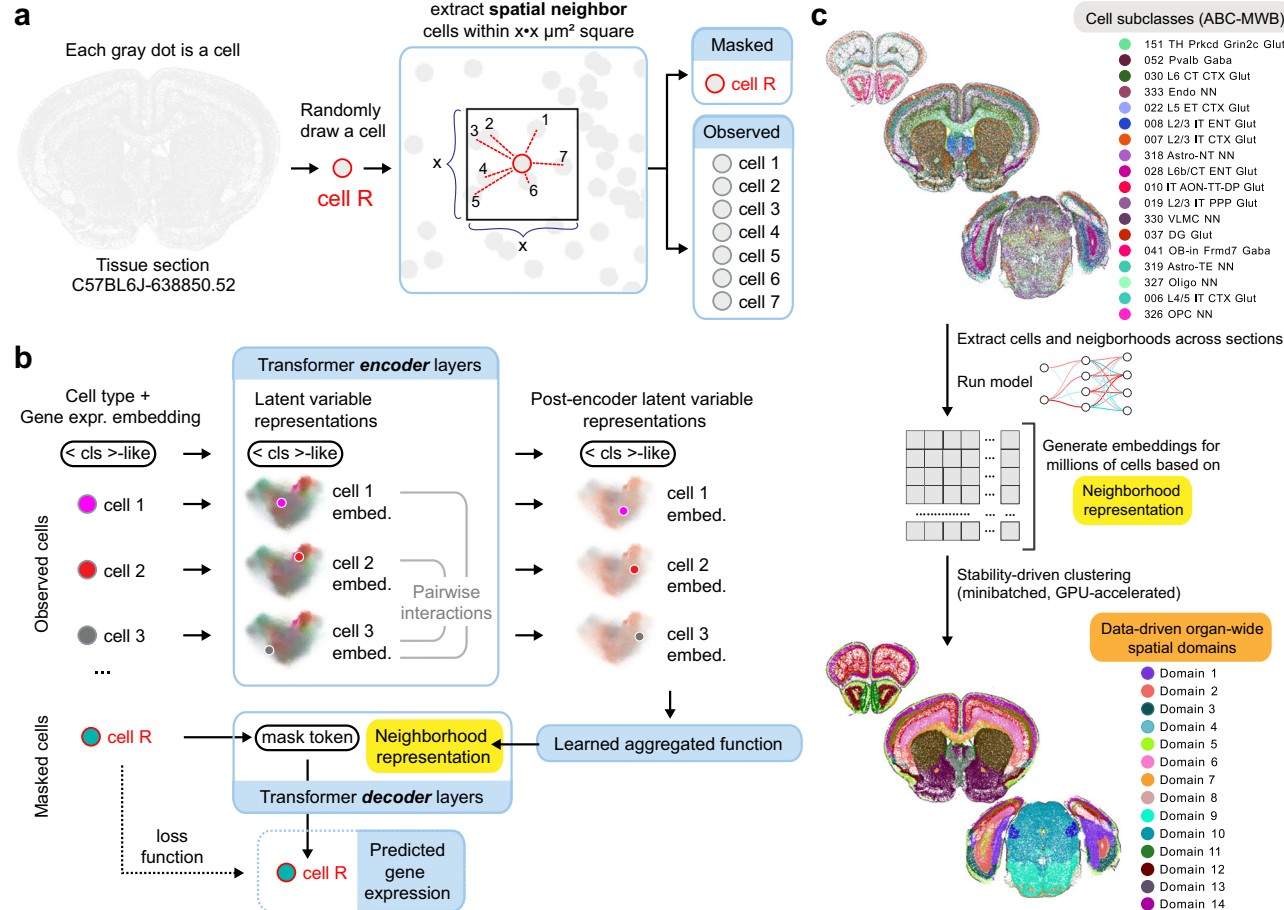

**Fig. 1 | Overall training and architectural scheme for CellTransformer. a** During training, a single cell is drawn (we denote this the reference cell, highlighted in red). We extract the reference cell's spatial neighbors and partition the group into a masked reference cell and its observed spatial neighbors. **b** Initially, the model encoder receives information about each cell and projects those features to *d*-dimensional latent variable space. Note that the latent variable point cloud shown is merely for illustration purposes and does not correspond to the actual CellTransformer latent space. Features interact across cells (tokens) through the self-attention mechanism. These per-cell representations and an extra token acting as a register token are then aggregated into a single vector representation, which we

refer to as the neighborhood representation. This representation is concatenated to a mask token which is cell type-specific and chosen to represent the type of the reference cell. A shallow transformer decoder (dotted lines) further refines these representations and then a linear projection is used to output parameters of a negative binomial distribution modeling of the MERFISH probe counts for the reference cell. **c** Once the model is trained, we compute embeddings (one for each neighborhood/reference-cell pairing) and concatenate these embeddings within the tissue section datasets and across tissue sections. Concatenating embeddings across tissue sections produces region discovery at organ level. We then cluster these embeddings using *k*-means to discover tissue domains across sections.

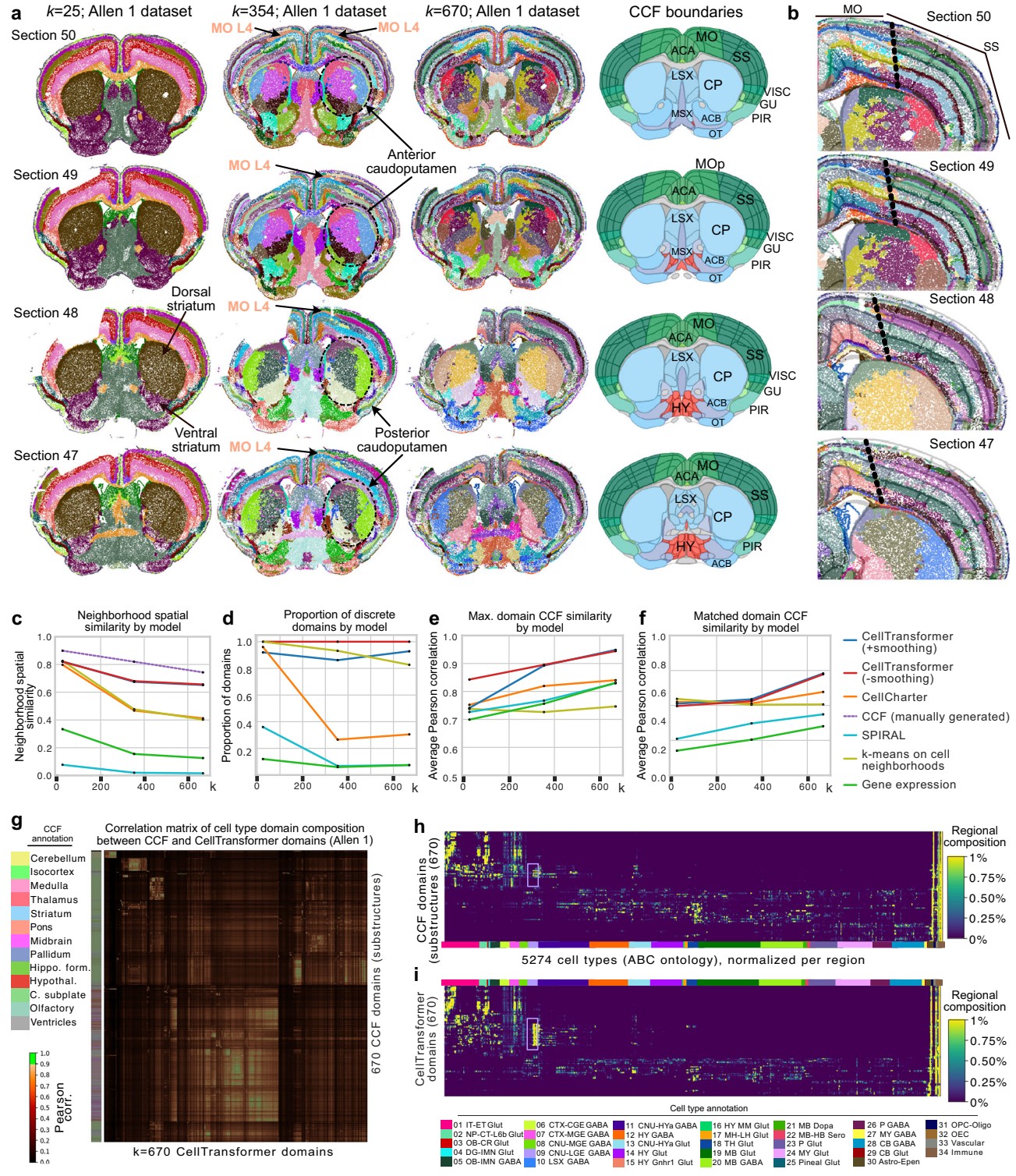

appear to have the highest magnitude in different cortical layers in the motor and visual areas, whereas PC3 has the highest magnitude in the lamina of the visceral and gustatory areas.

We generated domains at $k = 25$, 354, and 670, to match the division, structure, and substructure annotations' resolution in CCFv3, displaying domains for four consecutive tissue sections (Fig. 2a). We also provide representative images of spatial clusters across the brain (28/53 sections) at different $k$ in Supplementary Figs. 3–5. Low domain numbers such as $k = 25$ broadly divide the brain into neuroanatomically plausible patterns, with subregions of striatum (dorsal and ventral marked in Fig. 2a) and cortical layers clearly visible. A comparison of

cortical layers across these sections shows that CellTransformer domains at $k = 25$ are well matched to CCF (Supplementary Fig. 6b) and correctly identify major classes of layers (1, 2/3 4, 5, and 6) across somatosensory and somatomotor cortex. In particular, we point out the excellent correspondence of domains across tissue sections at $k = 25$ across the entire dataset (Supplementary Fig. 3), with nearly perfect consistency across regions. This suggested that our neighborhood representation method was robust enough to enable integration without modeling of batch or tissue-level covariates.

At $k = 354$, anterior-posterior subdivisions emerge such as the presence of layer 4 in the motor cortex (Fig. 2a, see Supplementary

**Fig. 2 | Representative images of spatial domains discovered using Cell-Transformer on the Allen 1 dataset (53 coronal sections and 500 gene MERFISH panel[1]) and comparison to CCF. a** Four sequential tissue sections (the inter-section distance is 200 μm) from anterior (first row, corresponding to section 50) to posterior (bottom row, section 47). In the first three columns, each dot is a cell, colored by spatial domain identified by CellTransformer when clustering was conducted with $k = 25$, 354, and 670 domains (the CCF division, structure, and substructure domain resolutions). Spatial domain labels are depicted with the same colors across sections within the same column. Fourth column shows CCF region registration to the same tissue section. Select regions are annotated with CCF labels. MO motor cortex, SS somatosensory cortex, ACA anterior cingulate, CP caudoputamen, LSX lateral septum, MSX medial septum, VISC visceral cortex, GU gustatory cortex, PIR piriform cortex, OT olfactory tubule, ACB nucleus accumbens, HY hypothalamus. **b** Single hemisphere images of same tissue sections in (**a**) domains fit at $k = 670$, zoomed in on cortical layers of motor cortex (MO) and somatosensory cortex (SS). CCF boundaries are shown in semi-transparent lines, with the boundary between SS and MO outlined in larger black dotted lines. **c** Spatial homogeneity (see *Methods*) of domains from different methods including recently published methods CellCharter and SPIRAL. **d** Proportion of discrete domains by model as compared to CCF at the same resolution. Single-cell level spatial smoothness was averaged over each domain instead of averaged over the

entire dataset as in (**c**). For each CCF annotation level we computed a threshold as the 20th percentile of per-domain spatial smoothness values. We applied this threshold to the distribution of data-driven domains at the same resolution. **e** Average Pearson correlation (averaging over number of domains and method) of the maximum Pearson correlation between the cell type composition (at subclass level, 338 types) vectors of data-driven regions with CCF ones. **f** Average Pearson correlation (averaging over number of domains and method) of optimal matched pairs between data-driven and CCF regions, where CCF regions are only allowed to pair with one data-driven region per comparison. Matches fit using linear programming. **g** Region-by-region Pearson correlation matrix comparing cell type composition vectors from 670 CCF regions (at substructure level) with 670 spatial domains from CellTransformer. The CCF regions are shown on the left with their structure annotations from CCF at division level on the side of the plot. Correlations above 0.9 are shown in bright green to assist in visualization. **h** Cell type (cluster level) by region matrix for 670 CCF regions at substructure level. **i** Cell type (cluster level) by region matrix for 670 CellTransformer regions. Rows are normalized to sum to 1 in both (**h**, **i**). Colors along x-axis in both (**g**, **h**) show cell class annotations from ABC-WMB cell type taxonomy at class level to allow for visualization of composition in terms of known types. Cell types in the "09 CNU-LGE GABA" class are boxed in purple in (**h**, **i**) matching their color in the legend. Rows of both (**h**, **i**) are grouped using clustering to produce approximately similar structure.

Fig. 6d, e). Historically, the mouse motor cortex was thought to lack a granular layer 4[18], however recently, MERFISH, transcriptomic and epigenomic studies have confirmed its existence[1,18,19]. At $k = 100$ and $k = 354$, we find a domain corresponding to Layer 4 in the somatosensory cortex which clearly extends to layer 4 in the motor cortex. We also provide a UMAP visualization of the CellTransformer embedding space with cluster labels from $k = 354$ clustering (Supplementary Fig. 7), demonstrating the observed domain spatial organization can also be recovered in a low-dimensional representation.

At $k = 670$, the cortical layers identified at lower resolution are further partitioned into superficial, intermediate, and deep strata within several layers. We visualize cortical layers across sections in depth (Fig. 2b), showing CellTransformer not only identifies fine superficial-deep structure within cortical layers but also preserves the boundary between somatosensory and motor cortex (marked in thick black dotted lines in Fig. 2b). Taken together these results showed that CellTransformer robustly describes previously known anatomical structures.

We also examined the caudoputamen at various choices of $k$. At $k = 25$, the caudoputamen is one domain, which separates into broad spatially contiguous domains at $k = 100$. Interestingly, at $k = 354$ and $k = 670$, we observe domains that intermingle in a grid-like pattern (Fig. 2a, Supplementary Fig. 8) that strongly resembles the Voronoi parcellation established in Hintiryan et al.[20] through systematic projection mapping to caudoputamen. Notably, CellTransformer also captures the quadrant pattern in intermediate caudoputamen (sections 52, 50, and 49 in Supplementary Fig. 8), which Hintiryan et al. attributed to the differences in subnetwork reorganization. The correspondence of our transcriptomic domains to the Hintiryan et al. results, which are exclusively based on projection mapping (non-transcriptomic data), suggests the biological relevance of our representation learning workflow.

We compared CellTransformer to several other workflows to capture spatial coherency and multiresolution neuroanatomical annotations in CCF at the division, structure, and substructure levels. For comparison, we used two recent methods, CellCharter[21] and SPIRAL[22] that are scalable to millions of cells as benchmarks. Cell-Charter builds spatially informed embeddings for domain detection by concatenating the embeddings across scales, followed by dimensionality reduction and batch correction, while SPIRAL uses graph neural networks for batch effect correction and integration across scales. Additionally, we implemented two machine learning baselines. Gene-expression based domain detection is employed for spatial

transcriptomics data analysis and has been shown to be highly successful for brain parecllation[23–25]. Therefore, we conducted $k$-means clustering on the single-cell MERFISH probe counts. We also employed $k$-means clustering on cellular neighborhoods (represented as cell type count vectors). Many other GPU-accelerated methods, such as scENVI[4], STACI[26], spaGCN[5], STAligner[6], STAGATE[27], or GraphST[28], cannot be run on datasets that contain millions of cells due to computational constraints (see *Methods*). One reason is that several of these methods require the instantiation of a dataset-wide pairwise distance matrix between all cells either on GPU or in RAM, which is a prohibitively large matrix (~60TB for ~4 M cells).

To quantify the spatial coherence of domains, we developed a single-cell level spatial homogeneity metric. For each cell, we identified its nearest 100 spatial neighbors. We then quantified the proportion of neighbor cells within the same spatial domain label as the starting cell (Fig. 2c). Ideally, we would expect a high proportion of neighbor cells to be in the same spatial domain as the starting cell. In this comparison of neighborhood spatial smoothness, CellTransformer outperforms CellCharter (58.2% better spatial coherence at 670 domains) and SPIRAL (4091.2%). CellTransformer also outperforms the machine learning baseline based on $k$-means clustering on cellular neighborhoods (61.9% better spatial coherence at $k = 670$), as well as the gene expression baseline (419.2% better spatial coherence at $k = 670$). We also compared versions of CellTransformer domains with and without the optional smoothing step. Both versions perform very similarly, indicating the CellTransformer representation efficiently suppresses high-frequency spatial features and changes relatively little after smoothing with a small bandwidth. For reference, we include the CCF parcellation (dashed purple line) in this comparison to provide an upper bound.

CellTransformer utilizes cell type information in both the encoder and decoder portions of the network. To understand the impact of this choice, we trained two additional variants of CellTransformer, one without cell type information in the decoder and one without cell type information in the encoder and the decoder (only expression). We generated spatial domains from these models' embeddings with smoothing, performed identically as in the base model. These variants perform competitively with the base CellTransformer version. The base model domains were 3.0% more similar to CCF than the model without cell type decoding, and 5.4% more similar than the model without cell type in either the encoder or decoder (Supplementary Fig. 9a). The smoothed and unsmoothed CellTransformer domains were also very similarly spatially smooth (Supplementary Fig. 9b), even

when the number of domains was extended to 2000. CellTransformer variants also produced relatively little decrease in spatial smoothness at greater than 1000 domains, whereas CellCharter smoothness sharply declined. Similarly to the CCF comparison, the model without cell type decoding performed better than the model without cell type in both the encoder and decoder. Taken together, these results indicate that cell type conditioning is an important component of the model that increases similarity to CCF and spatial uniformity. This may be because the cell types, which were fit using whole-transcriptome scRNA-seq profiling, implicitly comprise a whole-transcriptome imputation step. However, we also note that the CellTransformer variant without any cell type information still performs better than CellCharter with respect to similarity to CCF and spatial coherence.

In addition to dataset-wide spatial smoothness, we also quantified the proportion of discrete domains (Fig. 2d). To classify a given domain as spatially discrete, we first averaged the single-cell level smoothness values over domains. We then investigated the distribution of per-domain average smoothness (Supplementary Fig. 10a, b). CellTransformer's regional smoothness distribution is more similar to CCF than CellCharter, with relatively more highly smooth regions. Additionally, CellTransformer domains with smoothing are more similar to CCF than without. For a given CCF annotation resolution, we then computed a threshold based on the 20th percentile of per-domain averaged CCF smoothness values, which we used to classify a given data-driven domain as discrete or not. This adaptive metric allows us to compare fairly to human annotations at different resolutions. CellTransformer domains fit with and without smoothing both perform well, while for CellCharter, SPIRAL, and the gene expression baselines, discreteness declines significantly at 354 and 670 domains (Fig. 2d). We also computed the proportion of spatially discrete domains for resolutions from 700 to 2000 data-driven domains using the 20th percentile cutoff from the $k = 670$ CCF annotation level. For CellCharter, the proportion of discrete domains significantly diminishes (Supplementary Fig. 10c), unlike CellTransformer. The unsmoothed CellTransformer embedding workflow was most performant; we reasoned that the isotropic Gaussian smoothing employed may have eroded fine laminar boundaries, despite removing the isolated non-uniform domains discovered in striatum (Supplementary Note 1). We visualized a series of sections comparing domains from the smoothed and unsmoothed CellTransformer embeddings, finding a slight erosion of fine lamina in the cortex (Supplementary Fig. 11) compared with the unsmoothed, consistent with this observation. We also provide visualizations of a larger set of sections in Supplementary Fig. 12. To quantify the similarity of detected domains with CCF annotations, we compared the cell type composition of domains using cell type calls from the ABC-WMB taxonomy. We again chose the subclass cell type level, extracting for each domain and for each method a 338-long cell-type vector. We calculated the Pearson correlation of cell type composition vectors computed using the CCF regional annotations at division (25), structure (354) and substructure (670) levels against those of the various methods at the corresponding number of spatial domains. First, for each data-driven domain, we computed the maximum correlation to any CCF domain at the same CCF annotation resolution averaging these numbers across domains. CellTransformer outperforms other methods at mid-granularity and fine-granularity (Fig. 2e). In this comparison, several data-driven regions can match the same CCF region, which in the worst case could provide an overly optimistic picture of the correspondence between data-driven domains and CCF. To address this, we conducted a second analysis where only one CCF region could be matched to a given data-driven one. We used linear programming to compute an optimal 1:1 pairing of data-driven regions to CCF ones based on their Pearson correlation. We then averaged correlations across regions (Fig. 2f). CellTransformer is highly performant, showing that increase in correlation is not due to redundant matches to a single area in CCF.

Visualization of spatial clusters from CellCharter (Supplementary Figs. 13–14) at $k = 670$ domains across the brain and in midbrain shows lack of spatial coherence in cortical layers and midbrain, with detected domains distributed in a what appear to be non-biological patterns. This is particularly evident in thalamus and across cortical layers. In contrast, CellTransformer identified spatially coherent domains and uncovered plausible neuroanatomical structures.

Next, we directly compared CCF annotation domains with data-driven domains using normalized mutual information (NMI) and the adjusted Rand index (ARI) (Supplementary Fig. 15). CellTransformer (with and without smoothing) perform significantly better than comparator methods (13.4% higher NMI than CellCharter and 46.7% higher NMI than SPIRAL with smoothed embeddings, 13.5% higher NMI than CellCharter and 46.9% higher NMI than SPIRAL at 670 domains). Registering spatial transcriptomics data to CCF's dense MRIs is significantly challenging. We therefore attribute the overall low magnitude of the ARI and NMI, to potential errors in registration.

To further characterize the similarity of CellTransformer domains with CCF, we plotted the Pearson correlation matrix (Fig. 2g) between cell type composition vectors generated at 670 domains (substructure level in CCF). Block structures with very high correlations (>0.9, shown in bright green) in the matrix clearly show that CellTransformer is able to identify regions that are highly similar with known ones without any labels. We also investigated the correspondence of cell type composition with more granular single-cell annotations, employing the lowest-level single-cell annotations (the "cluster" level, with 5274 cell types, as opposed to the "subclass" level with 338) from ABC-WMB. We observed high similarity between the "substructure" CCF domain set (Fig. 2h) and 670 CellTransformer domains (Fig. 2i) with an average Pearson correlation of CellTransformer to CCF domains of 0.853. This shows that the high correspondence of CCF and CellTransformer is robust to cell type resolution at which comparison occurs. CellTransformer identified an increase in the number of domains containing the 09 CNU-LGE GABA class (striatal/pallidal GABAergic neurons from lateral ganglionic eminence compared with the 670 CCF substructures, shown in light purple box in Fig. 2h, i), potentially suggesting the presence of uncharacterized developmental populations.

The observation of hierarchical grouping of domains at different choices of $k$ (for example, delineation of cortical layers and sublayers with increasing number of domains) prompted us to develop a strategy to evaluate an optimal number of spatial domains based on two metrics. We implemented a previously published strategy[29] for *Drosophila* embryonic spatial gene expression and[30] for 3D spatial gene expression in the adult mouse brain to determine the optimal number of domains using a stability criterion. We reasoned that the optimal choice of spatial domain number would feature minimal variability across clustering runs. In brief, we computed 20 clustering instances with different random initializations for a large range $k$ values (100–2000) and quantified their variability over these initializations (see *Methods*). Interestingly, stability increased with increasing $k$ (Supplementary Fig. 16a, b). To facilitate the choice of a particular resolution for analysis, we also computed the inertia (sum of squared errors in embedding space) for each clustering solution. Low stability at small numbers of domains may partially explain subpar results for CellTransformer in the $k = 25$ CCF evaluations. We averaged the inertia curve and instability and computed the point of second derivative crossing to identify $k = 1300$ as our resolution for analysis (crossing point shown with red dot in Supplementary Fig. 16c).

CellTransformer is the only method out of the three deep learning methods we implemented (including six other pipelines, which were unable to cope with the size of the ABC-WMB dataset) to allow discovery of spatially coherent divisions at greater than CCF resolution. This study shows that spatial transcriptomics data can be used to identify brain regions at resolutions finer than previously defined in

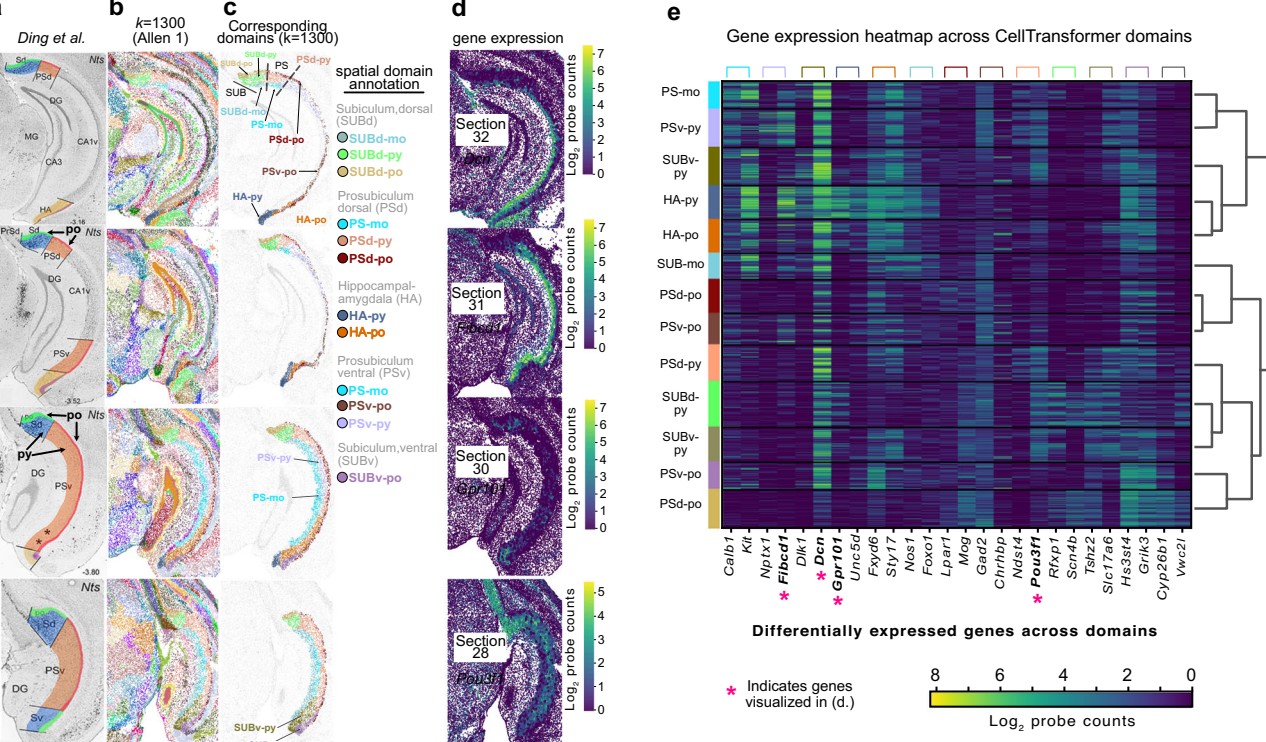

**Fig. 3 | Comparison of CellTransformer domain sets identified in the Allen 1[1] dataset with *k* = 1300 with a comprehensive region set found in Ding et al.[34], reproduced with permission from Elsevier. a** Representative images reproduced from Ding et al. of region boundaries in prosubiculum (PS), subiculum (SUB), and hippocampal-amygdala (HA) particularly along the dorsal-ventral axis. Polymorphic and pyramidal layers of the dorsal subiculum (SUBd) and ventral prosubiculum are indicated. **b** Images from hippocampal formation across 4 sequential tissue sections (anterior to posterior) roughly aligned to sections presented in Ding et al. Each dot is a cell colored using domain labels with *k* = 1300. **c** Same as (**b**) but only showing cells inside PS, SUB, and HA. Putative regional annotations are indicated and grouped by dorsal or ventral region within PS and SUB. **d** Gene expression patterns visualized at the corresponding tissue section, where only cells within PS/SUB/HA are shown. Units are in log₂ probe counts. **e** Gene expression heatmap of identified subregions, with putative anatomical annotation. Dendrogram from

hierarchical clustering in gene expression space is shown to the right. Genes visualized in (**d**) are bolded and denoted with a pink asterisk. Colored brackets indicate the genes which are differentially expressed with respect to the domain (colors match those shown on the left of the heatmap). Two genes per domain are shown and each gene is expressed with at least log-fold change greater than 1 relative to the other domains. Abbreviations: PS-mo prosubiculum molecular layer, PS-py pyramidal layer of subiculum, SUBv-py ventral subiculum, pyramidal layer, HA-py hippocampal-amygdaloid transition area, pyramidal layer, HA-po hippocampal-amygdaloid transition area, polymorphic layer, SUB-mo subiculum, molecular layer, PSd-po dorsal prosubiculum, polymorphic layer, PSv-po ventral prosubiculum, polymorphic layer, PSd-py dorsal prosubiculum, pyramidal layer, SUBd-py dorsal subiculum, pyramidal layer, SUBv-py ventral subiculum, pyramidal layer, PSv-po ventral prosubiculum, polymorphic layer, PSd-po dorsal prosubiculum, polymorphic layer.

the CCF. CellTransformer domains are more spatially smooth than comparator deep learning methods, as well as clustering on either gene expression or cell type composition, both at the single-cell level and at the domain level. Additionally, CellTransformer domains are more similar to CCF, both at the single-cell level, using NMI and ARI, and at the regional composition level. CellTransformer can also be used with and without cell type labels, although performance is unsurprisingly better when using cell type information. Crucially, CellTransformer is highly performant at discovering a high number of domains, whereas methods such as CellCharter experience a significant loss of spatial coherence (Supplementary Fig. 14). Encouraged by these findings, we next sought to establish correspondence of particular domains at *k* = 1300 to known neuroanatomy.

## Mapping of spatial domains in the hippocampal formation

We characterized CellTransformer's ability to capture known anatomical structure in the hippocampal formation, notably the subiculum (SUB) and prosubiculum (PS), in the Allen 1 dataset. We focused on this area because it is well characterized with respect to both connectivity[31] and transcriptomic composition[32,33]. These structures were investigated in Ding et al.[34], where the authors performed consensus clustering of glutamatergic neurons and subsequent ISH experiments were used to comprehensively map domains in dorsal subiculum (SUBd)

and dorsal and ventral prosubiculum (PSd and PSv). Specifically, this and other recent works have noted the extensive laminar organization (superficial layers to deeper layers), and the dorsal-ventral organization of the subiculum[35–37]. This organization has been attributed to distinct and correlated patterns of gene expression and connectivity.

We qualitatively compared spatial domains discovered by CellTransformer with *k* = 1300 to the anatomical borders identified in Ding et al. (Fig. 3a). The subiculum features a three-layer organization referred to as the molecular (mo) layer, the pyramidal cell (py) layer, and the polymorphic (po) cell layer. Figure 3a shows a diagram of SUB and PS regions based on Ding et al. with the pyramidal and polymorphic layers of SUB and PS annotated in bold black text. Figure 3b shows discovered spatial domains at *k* = 1300 across four sequential sections corresponding to those in Ding et al.[34]. A subset of domains corresponding to SUB and PS are shown in Fig. 3c along with putative regional annotations. CellTransformer identifies a three-layer organization in the dorsal subiculum corresponding to that in Ding et al. labeled SUBd-py (light green), SUBd-po (gold), and SUBd-mo (grayblue). CellTransformer also correctly splits the SUBd and PSd shown with black dotted lines on the image of section 32. Three-layer strata are also observed in PSd, although notably the pyramidal layer domain extends caudally, consistent with transcriptomic studies[31–33] of SUB architecture. For instance, our PSd-po domains (sections 31 and 30)

strongly resemble the HGEA layer 4 found in Bienkowski et al.[31]. Note that differences may arise between panels in Fig. 3a, c because of sectioning variability and lack of exact match between sections in ABC-WMB and the Ding et al. study. In addition to the aforementioned regions we also observe high agreement in areas such as in the hippocampus-amygdaloid transition area (HA) and ventral prosubiculum (PSv).

Ding et al.[34] observed differential projection topology in dorsal subiculum versus ventral prosubiculum. Correspondingly, genes were found to form opposing gradients across the length of subicular areas. Dorsolateral gene gradients appeared in SUBd and ventromedial gradients in PSv. Since CellTransformer domains appeared to correspond well with literature results, we explored gene expression patterns across domains to verify whether dorsal-ventral and medial-laterally varying gene patterns could be observed. We conducted differential expression analysis across our subicular domains (Fig. 3e), which, when visualized (Fig. 3d), clearly reflected these gradients. Many genes expressed in SUB and PS traverse their long axis as reported previously[32]. The identification of spatial domains that subdivided specific layers of PS and SUB, similar to the results in Ding et al. and featured similar types of gene expression gradients as existing literature, suggests that our pipeline was successful in learning neuroanatomically useful information. Importantly, while results in Ding et al. and related works were enabled by significant neuroanatomical and experimental expertise, CellTransformer allows identification of granular tissue structure in a data-driven fashion. Encouraged by this result, we continued our investigation of CellTransformer correspondence with known literature with a comparison in superior colliculus.

## CellTransformer enables quantification of laminar and columnar organization in the superior colliculus

Recent studies using systematic mapping of cortico-tectal fibers in the superior colliculus (SC) have identified distinct laminar and columnar structure[38], suggestive of the complex role SC plays in the integration of sensory information and the coordination of signals. Therefore, SC presented an excellent opportunity to identify transcriptomic and cellular correlates of connectomic variation. We observed a strong correspondence of three of our spatial clusters ($k = 1300$) in the Allen 1 dataset with known layers of superior colliculus, sensory area, particularly the zonal (zo), superficial gray (sg), and optic (op) layers across a set of tissue sections spanning ~600 μm from anterior to posterior (rows of Fig. 4a and Supplementary Fig. 17a). CellCharter was unable to identify these structures (Supplementary Fig. 14) at $k = 670$ (demonstrating a significant drop in spatial coherence at higher $k$) and only identifies two layers in SC, which does not conform with existing understanding.

By visualizing the cell type composition within the top-ten most abundant types for these three spatial domains (Supplementary Fig. 17a, b), we were able to identify cell types that were highly selective for our data-driven SC layers: types 0873 SCsg Gabrr2 Gaba_2, 0861 SCs Pax7 Nfia Gaba_3, and 0788 SCop Sln Glut_1. Crucially, the cell types, which have already been annotated as being associated with one of the zonal, optic, or superficial gray, are identified automatically by CellTransformer. We chose the supertype level to allow inspection of abundant cell types without being difficult to visualize. Supertype-level visualizations also show that even with granular cell types (1201 types in Yao et al.) CellTransformer domains are often marked by spatially specific cell type patterning; we note that we do not filter cells outside of our putative superior colliculus layers for visualization. Next we visualized the percentage of cells in each domain (Supplementary Fig. 17c), grouping them by neurotransmitter class (GABA-ergic, glutamatergic, and non-neuronal). The superficial gray layer showed the higher proportion of GABA-ergic neurons, while the optic layer had the highest proportion of glutamatergic neurons. To further explore these relationships, we calculated the number of distinct cell types

(supertype level) within each neurotransmitter class and domain. A clear dorsal-ventral organization was evident (Supplementary Fig. 17d) with the number of GABA-ergic and glutamatergic neuron types increasing with layer depth, suggesting CellTransformer's ability in capturing complex patterns of cellular spatial organization.

Encouraged by these findings, we also investigated subregions of the intermediate gray and intermediate white areas of the motor-related areas in SC (Fig. 4a, b), where we identify consistent regions across two consecutive sections that are not annotated in the CCF (rows of Fig. 4b). We define subregions of intermediate gray (ig) and white (iw), noting a medial-lateral structure similar to that in Benavidez et al.[38], which exhaustively cataloged projection zones in superior colliculus. Notably unlike in superior colliculus sensory, a significant number of non-neuronal cell types are found in very similar proportions across the intermediate white and gray layers (Fig. 4c), and instead differences in regions may be attributable to varying proportions of rare cell types. Encouragingly, even in these fine-grained areas, cell types that are highly specific for our data-driven layers can be readily identified (columns of Fig. 4b). These rare domain-enriched cell types include: 0849 SCm-PAG Cdh23 Gaba_2 (enriched in the medial intermediate white layer, shown in dark green), 0769 SCig SCiw Pitx2 Glut_3 (enriched in lateral intermediate white, shown in light blue), and 0764 SCig-an-PTT Foxb1 Glut_1 (enriched in medial intermediate gray, shown in dark blue). The identification of *Pitx2*-expressing neurons also supports our assertions that CellTransformer identifies biologically relevant domains, with previous studies using *Pitx2* expression specifically as an intermediate layer marker in superior colliculus[19,35].

We observed complex cell type abundance gradients when visualizing the percentage of cells in a given domain by their neurotransmitter type (Fig. 4). We used supertype level to confirm that spatially-varying cell distribution patterns persisted when using more granular cell type annotations. Lateral domains such as intermediate gray, lateral (shown in dark blue) and intermediate white, lateral (shown in light blue) featured a smaller proportion of GABA-ergic neurons than medial domains but were enriched for glutamatergic neurons and non-neurons (Fig. 4d). Despite the low proportion of GABA-ergic neurons, the lateral domain of intermediate gray possessed the highest number of GABA-ergic neuron cell types (Fig. 4e). Conversely, non-neurons featured the same number of types across the laminae.

## A medial-lateral gradient of inhibitory neurons in the midbrain reticular nucleus

Next, we investigated the midbrain reticular nucleus (MRN), a subcortical structure with few anatomical annotations in CCF. MRN is highly enriched for interneurons and appears to play a complex role in movement initiation and release[36,37]. CellTransformer identifies four subregions of the MRN, which are not included in the existing CCF annotation (Fig. 5a). Plotting cell type proportions across the MRN, we identified cell types that are enriched for these putatively uncharacterized areas, although all domains were predominantly glial (e.g., 1184 MOL NN_4 supertype is abundant in all regions, Fig. 5b). Interestingly, several neuronal types found in these subregions were originally annotated as belonging in inferior colliculus (i.e., 0811 IC Six3 En2 Gaba_5 and 0809 IC En2 Gaba_3)[1]. However, the fine-grained region derivation by CellTransformer prompted us to perform additional visual inspection (see CCF annotation of MRN identified in Fig. 5a with black arrows) of the MERFISH data and conclude that these neurons are clearly located in MRN. In addition, by visualizing differentially expressed genes across the domains (Fig. 5c), we identified genes that were subregion selective (selected genes shown in Fig. 5d) and form dorsal-ventral expression gradients. Hierarchical clustering showed that the two dorsal domains (purple and brown) group together with the two ventral ones (gold and gray).

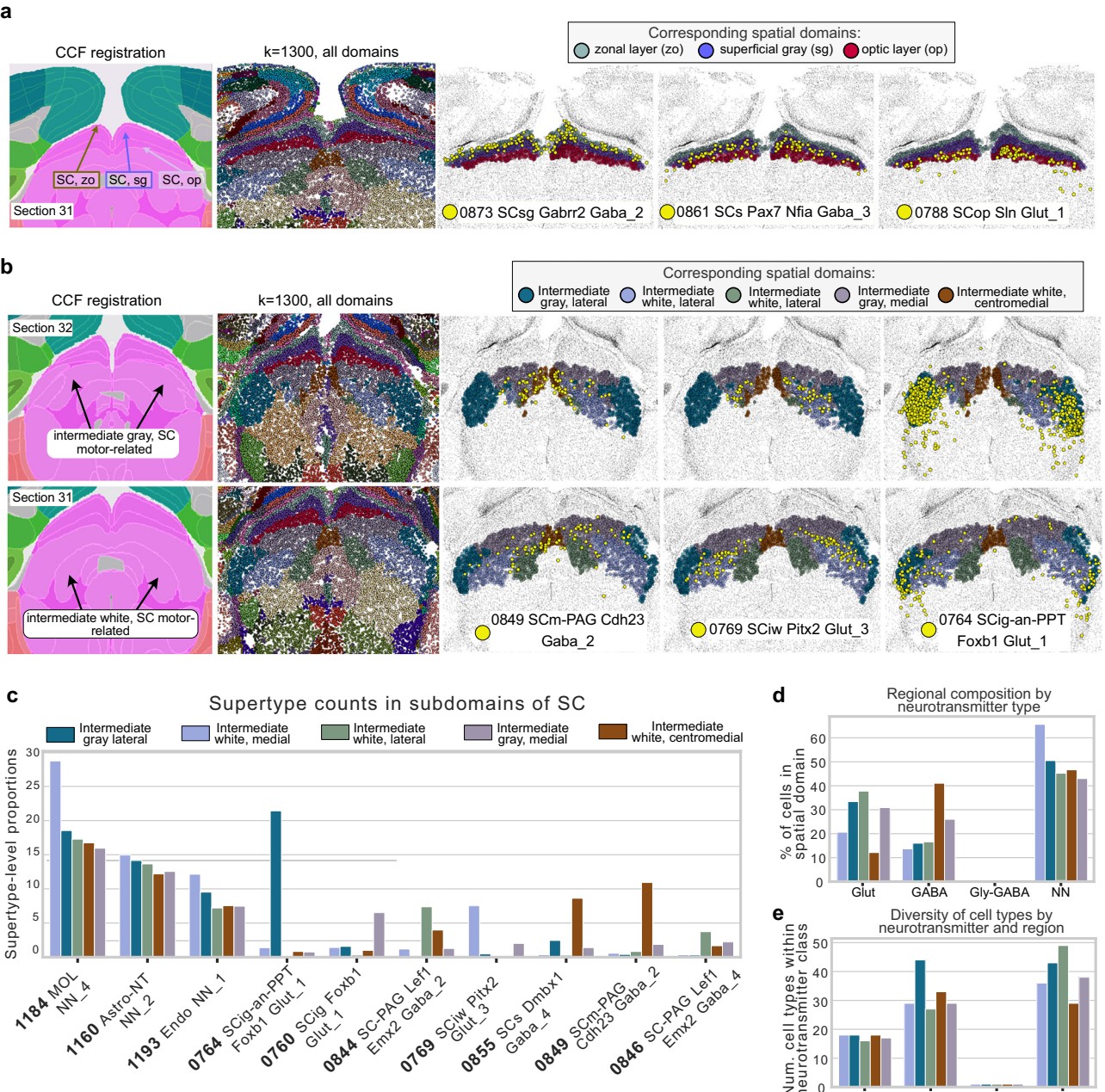

**Fig. 4 | Examination of putative dorsal and ventral subregions of superior colliculus identified in the Allen 1[1] dataset using CellTransformer. a** Putative subregions of sensory layers of superior colliculus in tissue section 32 identified with $k = 1300$ CellTransformer domains. CCF registration is in the first column, with zonal (zo), superficial gray (sg), and optic (op) layers labeled by the color of their CellTransformer domain in third, fourth, and fifth columns. The second column shows all cells with color labels from their spatial domain from CellTransformer at $k = 1300$. The third, fourth, and fifth columns show the putative zo (gray-green), sg (purple), and op (red) domains. These columns also show the spatial distribution of one supertype level cell type in yellow across the section. **b** Sequential tissue

sections (32: anterior, 31: posterior) shown similarly to (**a**) but visualizing subregions of the intermediate gray and intermediate white layers, which are indicated with black arrows in the CCF registered annotation image. **c** Proportions of different supertype level cell types for top-ten most abundant types in different spatial domains. Colors refer to the same spatial domain label in (**a**, **b**). Cell types visualized in (**b**) are denoted with a yellow asterisk. **d** Barplot of the percentage of cells of a given neurotransmitter class found in a given region (GABA GABAergic, Glut Glutamatergic, NN non-neuronal). **e** Number of unique cell types (at supertype level) found in each domain, grouped by neurotransmitter class.

We again visualized the neurotransmitter composition and the number of unique cell types of given neurotransmitter classes. We observed that the number of types of excitatory neurons was spatially graded. Dorsal domains of MRN (domain 365 shown in purple and domain 198 shown in brown, (Fig. 5e) featured the highest proportion of glutamatergic neurons, and the proportion of glutamatergic neurons in MRN domains decreased with increasing depth. Non-neuronal cells are also organized along this gradient, but in the

opposite direction, the ventral areas featuring the highest proportion of glia and the dorsal areas the lowest. Interestingly, MRN domains composed of a higher proportion of glutamatergic neurons were also the ones with the greatest number of glutamatergic neuron types, also following the dorsal-ventral gradient (Pearson correlation $r = 0.89$). This relationship was observed for nonneuronal cells (Pearson correlation $r = 0.81$, Fig. 5f), but not for GABA-ergic neurons (Pearson correlation $r = -0.64$). This suggests that CellTransformer

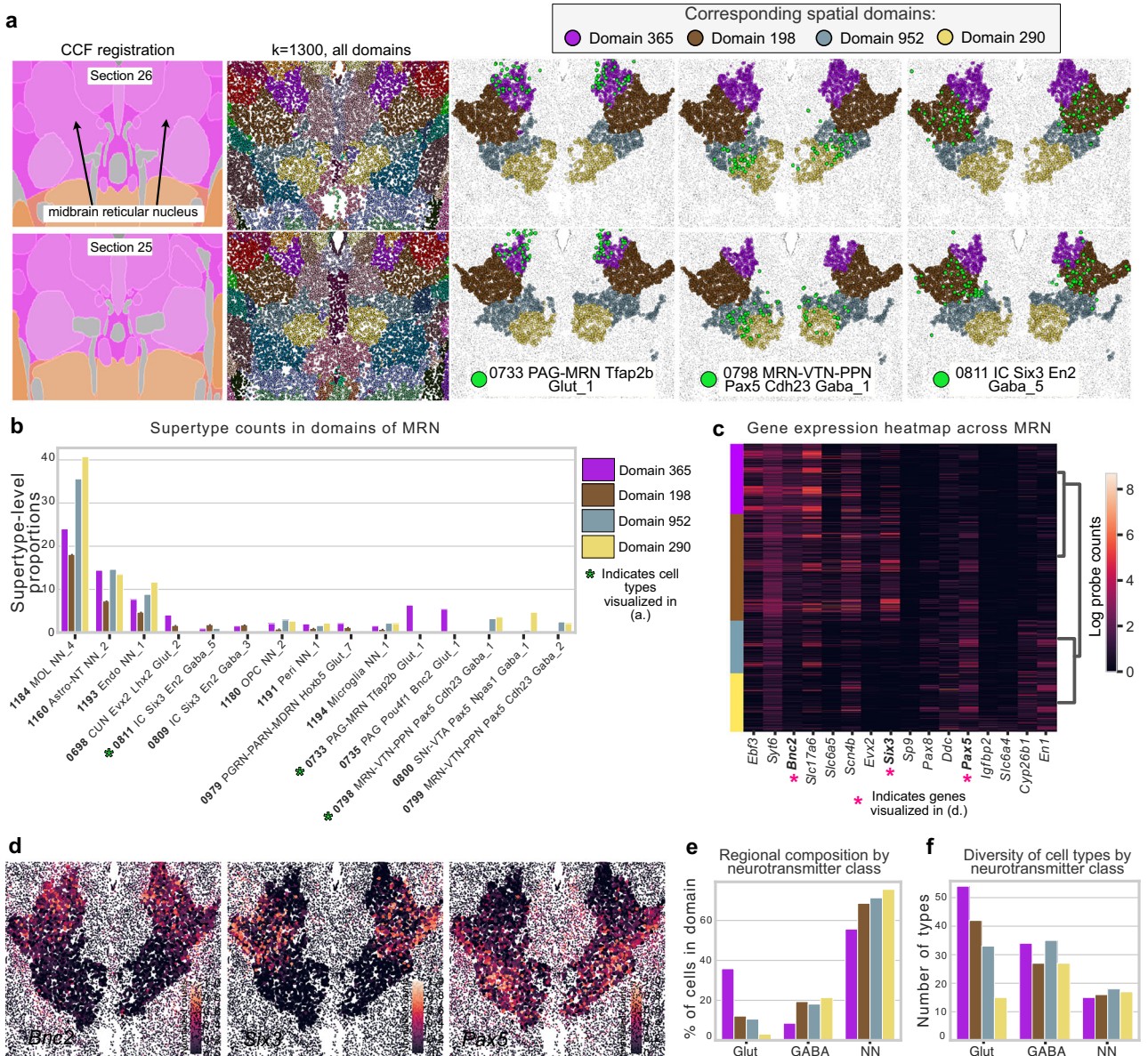

**Fig. 5 | Subregions of midbrain reticular nucleus discovered in the Allen 1[1] dataset using CellTransformer. a** Sequential tissue sections (26 and 25, anterior to posterior). First column: CCF registration with approximate location of midbrain reticular nucleus shown with arrows. Note that registration is not exact and can differ across hemispheres. Second column: all cells in field of view, with color from spatial domains determined with $k = 1300$. The rest of the columns show only cells located in the MRN, and each column shows a different supertype level cell type in green. **b** Supertype level cell type proportions for top fifteen most abundant types across the MRN subregions. Cell types visualized in (**a**) are denoted with green asterisks. **c** Selected 4 differentially expressed genes across regions. Each gene is expressed at least log fold change greater than 1 relative to the other domains. MERFISH probe distributions for select genes indicated with pink asterisks are shown in (**d**). **d** Gene expression gradients across tissue section 25 for *Bnc2*, *Six3*, and *Pax5*, showing specificity for each of the putative MRN subregions. Intensity of color is 0–1 normalized after log scaling raw probe counts. Each dot is a cell, and the color shows the relative transcript count. We show only cells within the subregions to make it visually easier to distinguish the relevant cells. **e** Bar plot of the percentage of cells for a given neurotransmitter type found in each domain, (GABA GABAergic, Glut Glutamatergic, NN non-neuronal). **f** Number of unique cell types (at supertype level) found in each domain, grouped by neurotransmitter class.

can identify plausible structures even in historically difficult to characterize areas.

## CellTransformer enables scaling up to multi-animal, million cell datasets and generalizes to other spatial assays

In order to investigate CellTransformer's ability to integrate across animals, we trained a new model from scratch on the Zhang et al.[12] MERFISH data, which uses an 1129 gene panel and is split over four animals, with both coronal (Zhuang 1 and 2) and sagittal sections (Zhuang 3 and 4). We computed embeddings for each neighborhood as in the previous analysis and performed $k$-means clustering,

concatenating representations for all mice and sections. This provided an opportunity to examine whether CellTransformer could adapt to a multi-animal case in addition to finding spatial domains across tissue sections of the same animal.

Spatial domains in sequential tissue sections appeared highly concordant across all four mice (Fig. 6a) at the 50-domain resolution. We used 50 domains to facilitate clear visualization of the domains across animals with relatively few colors. Coronal and sagittal sections across mice clearly corresponded anatomically. Cortical layers were highly consistent across animal and section orientation. Structures that appear in the coronal view can be readily identified in the sagittal

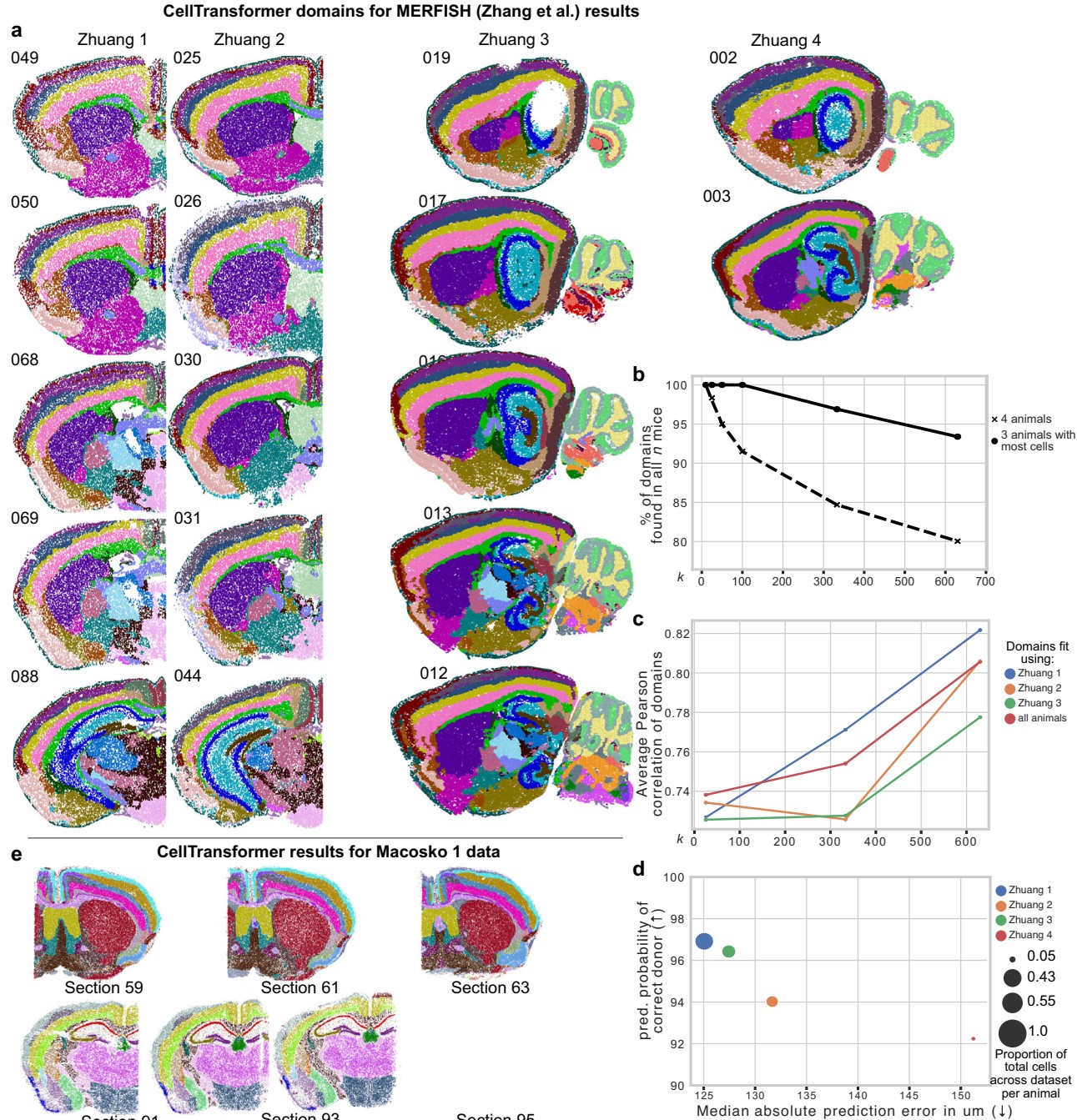

**Fig. 6 | Investigation into performance of CellTransformer on the Zhuang 1-4 datasets (239 sections, both coronal and sagittal, with a 1129 gene MERFISH panel[12]). a** Representative images of all four mice arranged by column. The section number for each mouse is shown in the upper left of each image. Note that Zhuang 4 only had three sagittal sections, which only correspond to one hemisphere of the brain. For each image, each dot is a cell neighborhood and colors come from a spatial clustering with $k = 50$ (number of CCF regions at structure level), fit by concatenating embeddings across mice. **b** Quantification of number of per-mouse specific spatial clusters, computed by clustering at different $k$ and computing the number of clusters found for all mice (4 animals) and for the three mice with the most cells per mouse (Zhuang 1, 2, and 3). Note that because serial sections were collected at a higher frequency (100 μm versus 200 μm), different areas of the brain will have marginally higher coverage in one brain or another. This is particularly relevant for Zhuang 4, for which sections only correspond to one hemisphere of the brain and therefore will have limited correspondence with domains in the other animals, which either cover the entire brain anterior-posterior (Zhuang 1 and 2) or across the sagittal axis (Zhuang 3). In contrast, Zhuang 3 does not fully cover a single hemisphere. **c** Average correlation of the cell type composition of brain regions computed CellTransformer to CCF regions, computed using the linear-sum assignment matching algorithm (exclusively matching regions from one set to the other). Dotted lines with "o" marker indicate results when fitting using all three mice with >1 M cells together. Solid lines with "x" marker indicate results when computing spatial clustering on each mouse in isolation. **d** Quantification of subject-level information present in embeddings using linear regression. The median absolute prediction error (x-axis) quantifies accuracy in predicting the (x, y, z) coordinates of a neighborhood from its embeddings. The y-axis quantifies accuracy when predicting mouse identity from embeddings using logistic regression. Values are averaged across cells per mouse. **e** Results of domain discovery ($k = 50$) on a Slide-SeqV2 whole mouse brain (Macosko 1) dataset[13]. Two sets of three sequential sections are shown.

sections. For example the hippocampal formation (blue) is well delineated in sections 088 for Zhuang 1, section 044 for Zhuang 2, and across displayed sections of Zhuang 3 and Zhuang 4. Despite a relatively low number of cells in mouse 4 (162,579 cells versus more than 1.0 million for each of the other animals), nearly all spatial domains observed for Zhuang 4 are present in other animals. Note that sections from this animal only cover a section of the lateral portion of the brain and do not span the entirety of the sagittal plane.

We quantified the robustness of CellTransformer domains in a multi-animal context across and within Zhuang 1-4 datasets. We ran clustering and identified domains at the three values of $k$: 25, 333, 630. These $k$ values correspond to three CCF resolution levels reported by registration in Zhang et al. (note the number of domains differs due to registration differences). For each $k$ value, we counted the number of domains observed in all four animals. We also repeated this analysis without data for Zhuang 4, which contains far fewer cells than the datasets from other animals (Fig. 6b). We find that even at high resolution (630 domains), 93.3% domains were found in each mouse, showing high consistency of CellTransformer domains across datasets. With the Zhuang 4 included, at 630 domains, 80.0% domains were found in every animal. To verify that domain consistency across animals was not related to loss of domain spatial coherence, we repeated the neighborhood smoothness analysis we developed for analysis of the Allen 1 dataset on the combined Zhang et al. data. Spatial smoothness was similar to that of Allen 1 (Supplementary Fig. 18), indicating CellTransformer can discover spatially coherent domains that are robustly integrated across animals.

We next quantified the similarity of CellTransformer domains to CCF regions. Similarly to our analysis of the Yao et al. dataset, we computed average similarity of cell type composition vectors from CCF and CellTransformer. In domain discovery across all animals, we found cell-type composition vectors that correspond strongly to CCF (Pearson correlation = 0.805, red line in Fig. 6b). We also evaluated whether clustering only on embeddings from one animal would significantly affect similarity to CCF. Correspondence between CellTransformer domains and CCF is high even when domains are fit with a subset of the dataset (Pearson correlations >0.7 for all comparisons across resolutions and domain source, Fig. 6c). This demonstrates CellTransformer can reproduce a consistent neuroanatomical structure even with a small number of observations. Results were highly similar overall to CCF in the Zhang dataset and Yao dataset (Pearson correlations greater than 0.6 for all comparisons), indicating CellTransformer's robustness to changes in gene panel and preprocessing choices.

To further investigate how donor metadata was encoded in the embeddings, we employed linear probing strategies commonly used in interpretation of deep learning model embeddings. We regressed CCF-registered (x, y, z) coordinate position across all embeddings and used logistic regression to classify animal identities. Neighborhood level prediction of donor identity was very accurate (>94% for all animals, Fig. 6d) and median absolute prediction error was accurate within 151 μm. Decoding accuracy for both metrics was strongly associated with the number of cells (Pearson correlation −0.92 for coordinate error and 0.92 for predicted donor probability). The observation that mouse donor identity is easily predicted from per-neighborhood embeddings while still maintaining cross-animal and cross-section coherence is another demonstration of the richness of the representation learned by our approach.

Finally to demonstrate the applicability of our strategy to a different spatial transcriptomics modality, we analyzed a whole mouse brain Slide-SeqV2 dataset ("Macosko 1"), collected in Langlieb et al.[13]. Slide-SeqV2 provides whole transcriptome coverage in a spatial context by tiling tissue slices with 10 μm by 10 μm squares. As each square may contain more than one cell or a partial cell, we fit our model to the deconvoluted single-cell data computed using the RCTD[39] method

provided. This produced 4,783,976 cells across 101 slices. We also filtered the dataset for low-quality cells and infrequently expressed genes (see *Methods*). We found that increasing the size of our model (from 4 encoder layers to 10) was necessary to identify spatially coherent domains, perhaps driven by the much larger number of genes detected (5019 versus 500 or 1129 in the two MERFISH datasets; see *Methods*). We plot three sequential sections from domain discovery at $k = 50$ (Fig. 6e). We show that CellTransformer robustly identifies cortical layers across sections and in known structures such as the midbrain and the piriform areas. Domain discovery with greater than 50 regions did not produce adequate integration across sections, possibly because of variable cellular density and single-cell read depth across sections.

Overall, we found that the CellTransformer workflow successfully identifies interpretable domains across different spatial transcriptomics modalities, and that the resolution of cross-section and cross-dataset domains depends on the specific spatial transcriptomics method and the quality of the datasets.

## Discussion

In this study, we present a transformer-based pipeline to combine scRNA-seq and spatially resolved transcriptomic atlases to perform accurate organ-level domain discovery. We developed a custom representation learning workflow and implemented a computationally efficient pipeline that readily allows scaling to multi-million cell, multi-animal datasets. The representations learned in our model can be clustered to identify progressively finer-scale spatial domains directly from local cellular and molecular information alone, without pre-defined spatial labels. We show these regions can be interpreted at the gene or cell level and recapitulate a variety of existing findings in the neuroscience literature, where many existing methods cannot. Our pipeline allows the extraction of a very high number of domains, which retain high correspondence to the existing brain region ontology (CCF). These domains are also highly spatially consistent both within and across tissue sections and even over multiple animals. Not only can CellTransformer discover this fine structure, but it can reliably find it across animals, even with hundreds of regions. This capability is intrinsic to our model and is learned despite any conditional modeling for donor or section-level covariates, indicating the robustness of learned features.

We note that our objective is not to suggest that the domains discovered by our method are a definitive, normative set of brain regions, and we did not conduct any further validation studies to support these claims. Nor is it to strongly assert that the brain is composed of discrete brain regions, as opposed to a composition of gene expression gradients. Our objective was to develop a tool that would operationally identify plausible brain structures and sub-structures in a data-driven way and to satisfy a neuroanatomical convention for discrete domains. To support this goal, we used $k$-means clustering due to its capability to be mini-batched during fitting and because it produced biologically plausible domains; a probabilistic framework such as Gaussian mixture modeling may also have worked well, and possibly would have produced similarly plausible domains. Indeed, the fact that no finite number of domains was identified via our stability criterion suggests the possibility of a significantly larger number of domains.

We demonstrated the robustness of our model in uncovering biologically relevant models and characterized our pipeline's ability to reproduce known neuroanatomy in the hippocampal formation and superior colliculus. Detected domains were concordant with previous comprehensive transcriptomic and connectivity studies of these areas, but were identified in a scalable and data-driven way. Not only were we able to detect known and novel regions, we found that CellTransformer domains can recapitulate and extend known spatial cell type enrichment patterns and gene expression gradients.

We highlight several advantages of our architecture and approach. Although the use of graph-structured architecture or self-attention to model cells in a neighborhood graph is not novel[17,27,40], our approach is distinguished by a self-supervised training objective based on spatial correlation between a cell and its neighbors that facilitates learning of a fixed representation for a cellular neighborhood (Supplementary Note 2). The intersection of graph neural networks, transformers, and representation learning research is a rich and rapidly moving research area. Methods for spatial-graph structured data such as Cell-Transformer will benefit immensely from implementing more effective ways of encoding the data and its metadata such as better position encoding mechanisms[41], rotationally-invariant architectures[42], or arbitrary numbers of genes[43]. There are also significant opportunities to extend CellTransformer's local representation framework to include other data modalities. Using a transformer rather than a graph neural network facilitates the inclusion of arbitrary contextual data, such as cell-level (e.g., neurophysiology[44,45]) and pixel-level data (e.g., mesoscale axonal connectivity[46], or magnetic resonance imaging[47]), which can be tokenized and included in our framework.

Caveats of our approach include the necessity of a user-specified spatial radius (for neighborhood computation) and the choice of $k$ for spatial cluster detection. The stability of detected spatial domains at a given radius or $k$ poses an interesting future angle from which to study anatomical hierarchies in the brain. Users must also have access to GPUs (to allow for timely model fitting), which reduces overall accessibility, although the hardware requirements are still much less intensive than for many existing models such as spaGCN and scENVI[4].

CellTransformer advances the state of the art for automated domain detection by facilitating the identification of granular and biologically relevant spatial domains that are extensible to very large, multi-animal spatial transcriptomic datasets. As spatially resolved transcriptomic and multi-omics studies of the brain become more prevalent, tools such as CellTransformer provide avenues to transform data into refined anatomical maps of the brain and other complex organs and pave the way towards tissue-level structure-function mapping.

# Methods

## Allen Brain Cell Mouse Whole Brain (ABC-WMB) dataset processing

**Allen Institute for Brain Science dataset preprocessing.** We downloaded the log-transformed MERFISH probe counts and metadata for the Allen Institute for Brain Science animal ("Allen 1") from the Allen Institute public release (https://alleninstitute.github.io/abc_atlas_access/intro.html) access for ABC-WMB. The Allen 1 dataset is composed of 53 coronal sections. The MERFISH probe set included 500 genes. Serial sections were collected at 200 μm intervals. We used the taxonomy from the "20231215" data release. Allen 1 is composed of 3,737,550 cells. We transformed the $(x, y)$ coordinates of each cell into microns instead of mm as provided. Otherwise, the dataset was used as-is for neural network training.

**Zhuang lab (Zhang et al.) dataset processing.** Data were downloaded from the "20230830" data release from the Allen Institute ABC-WMB public data release. Two animals ("Zhuang 1" and "Zhuang 2") were collected with coronal sections. The other two animals ("Zhuang 3" and "Zhuang 4") were collected sagittally. Serial sections for Zhuang 1 (female) were collected at 100 μm intervals, while serial sections for the other animals (all male) were collected at 200 μm intervals. The size of the MERFISH probe set included 1129 genes. Zhuang 1 and Zhuang 2 consist of 2,846,909 cells and 1,227,409 cells, respectively. Zhuang 3 and Zhuang 4 consist of 1,585,844 cells and 162,579 cells, respectively. We transformed the $(x, y)$ coordinates of each cell into microns instead of mm as provided. Otherwise, the data were used as-is for neural network training.

**Cellular neighborhood construction.** We consider cells in the same neighborhood as a reference cell if the distance between them is within a box of fixed size. For all MERFISH datasets, we used a box width of 85 μm.

**CellTransformer architecture.** We construct a CellTransformer to generate a latent representation from a cellular neighborhood where this representation is composed of both molecular and cell type information. We represent cells as nodes in an undirected graph, $G = (V, E)$ where $V$ indexes the nodes in the graph (cells) and we add an edge $(x_i, x_j)$ to the edge set $E$ if $r > d_{i,j}$, with $r$ a user-specified distance in microns. We assume also that for each node we have access to $x_i \in R^g$, a $g$-dimensional vector of MERFISH probe or cell deconvoluted transcript counts. We also assume we are given class labels $\mathbf{c} = \{c_i \in \{1, \ldots, C\}\}_{i=1}^N$ for each of the $N$ cells. The user must also specify an embedding dimension and number of transformer encoder and decoder layers; in all experiments in this paper, using MERFISH data, we use an embedding dimension of 384, 4 encoder layers, and 4 decoder layers. For Slide-SeqV2 analysis, we used 10 encoder layers and 4 decoder layers.

To generate a neighborhood embedding, we identify a particular cell which we call a reference cell. Its first degree neighbors are extracted from $G$. We first apply a shallow encoder (two layer perceptron with GELU nonlinearity) function $f_\theta : R^g \rightarrow R^{192}$ which maps the gene expression into embedding space. We likewise construct and apply the function $g_\theta : R^C \rightarrow R^{192}$ to map one-hot encoded cell type labels to an embedding space, here a simple lookup into a learnable embedding table. These representations are concatenated into a single 384-dimensional representation. We apply these transformations to each cell in the neighborhood, not including the reference cell. We note that at this point, all operations have been performed per cell without interactions. In addition to these cell tokens, we also instantiate for each neighborhood a register token, which we use to accumulate global information across the neighborhood. We refer to this token as a <cls>-like token in keeping with previous literature.

We then apply a transformer encoder to the cells, only allowing cells within the same neighborhood and their <cls>-like tokens to attend to each other. We use 8 attention heads with GELU activations and layer norm prior to attention and MLP projection. We note that including a bias term in the key, query, and value MLPs is important to stabilize training, while not noting any significant differences in models fit with and without bias terms for the rest of the encoder and decoder layer MLPs. Following the transformer encoder, we use attention pooling to aggregate the cell and <cls> representations for each neighborhood into a single token with embedding dimension 384. We refer to these as the neighborhood representations.

We then instantiate a new token from each reference cell that is a learned embedding for each cell type (separately from the encoder cell type embedding). These are concatenated to the neighborhood representations. We then apply a transformer decoder to the tokens, allowing only the neighborhood token and masked cell embedding to attend to each other if they are from the same cellular neighborhood. This decoder embedding dimension was 384 with 8 attention heads.

During training, we extract only the masked reference cell tokens. We then use separate linear projections to output the mean, dispersion, scale, and zero inflation logit parameters for zero-inflated negative binomial regression. We optimize the model by minimizing the log likelihood of a negative binomial distribution using observed cells' MERFISH probe counts. We trained all versions of CellTransformer on a system with 2 NVIDIA A6000 GPUs with an effective batch size of 256.

**Spatial domain detection.** Once trained, we apply CellTransformer to a given dataset and instead of extracting reference cell tokens we extract the neighborhood representation. We then cluster this representation using $k$-means. We use the cuml library to perform this

operation on GPU (cuml.KMeans), with arguments n_init=3, over-sampling_factor=3, and max_iter=1000.

**Optional smoothing of embeddings.** We observe that spatial domains are spatially smooth. However, in the case that there is a high-frequency signal that the end-user would like to filter, we optionally introduce a step prior to $k$-means where we smooth the embeddings using a Gaussian filter. For all comparisons except those in Supplementary Fig. 12, smoothing was performed with a Gaussian filter with a 40 μm full-width at half maxima (sigma of 12.01 μm).

**Model fitting on the Allen 1 dataset.** We used an 80–20% train-test split proportion (random splitting across the entire dataset) and the ADAM optimizer over 40 epochs. We perform a linear warmup for 500 steps to a peak learning rate of 0.001 and use an inverse-square root learning rate scheduler to decay the learning rate continuously. We use a weight decay value of 0.00005, which we do not warm up.

**Model fitting on Zhuang datasets.** We perform training from scratch without transfer. We trained for 40 epochs with the same settings as for the Allen 1, with the exception of adapting projections to 1129 genes instead of 500.

**Computation of stability criterion.** We follow Wu et al. in using an Amari-type distance to compare clustering solutions. Briefly, we compute several replicates (20 in this work) of $k$-means at a given choice of $k$ with different random seeds $D_{k,i}$, with $i$ indexing the different centroids for a given solution. We then measure the stability of a given choice of $k$ by comparing the similarity of all pairs of $D_k$. Define $C$ the Pearson correlation matrix between pairs $D$ and $D'$. Then we use this dissimilarity metric:

$$\text{diss}(D, D') = \frac{1}{2K}\left(2K - \sum_{i=1}^{K} \max_{1 \le k \le K} C_{k,j} - \sum_{j=1}^{K} \max_{1 \le k \le K} C_{k,j}\right) \quad (1)$$

to identify the choice of $k$ which is most stable.

**Regional matching with CCF computation.** To quantify the overall similarity of regions extracted using CellTransformer with CCF, we first extract cell type composition vectors for each region at a given level of the hierarchy. For all comparisons in Fig. 2, we use the subclass level (338 cell types), resulting in $k$-region by 338 matrices. For each region derived from one of the tested models, we compute two quantities: the best match (maximum value of Pearson correlation, non-exclusively) to any CCF (Fig. 2d) or an exclusive match (using the linear sum assignment algorithm) to pair the regions from either set one-to-one (Fig. 2e). We then computed the average Pearson correlation across the paired matches as the metric. We use scipy.optimize's implementation to solve the linear sum problem.

**CellCharter.** To run CellCharter, we first generated scVI embeddings using the default settings for depth and width of the network and with the tissue section labels as conditional batch variables. We trained for 50 epochs using the early_stopping=True setting. We then aggregated across 3 (default settings), 6, 9 layers using the cellcharter.gr.aggregate_neighbors function. We then applied CellCharter's Gaussian mixture model implementation at various choices of the number of Gaussians. We could not run the mixture model with our hardware (A6000, 48GB GPU memory) for more than 9 layers, which was also the number that produced the highest correspondence with CCF and is reported in Fig. 2e.

**SPIRAL.** To run SPIRAL we generated edge sets for 40 μm, 85 μm, and 170 μm neighborhood radii. SPIRAL requires supervision on single-cell types so for this we use the subclass cell type levels. We trained models across neighborhood sizes for 1 epoch and then chose the neighborhood size with best performance (170 μm) and trained this model to saturation (10 epochs). SPIRAL uses four objective functions so to assess saturation we averaged them. We note that SPIRAL does not use a training and testing set split in their training, making it difficult to assess an optimal stopping point. For the $k = 354$ and $k = 670$ domain discovery analyses the SPIRAL clustering pipeline produced an out-of-memory error and we instead used our own pipeline with $k$-means on SPIRAL embeddings.

**Nearest-neighbor smoothness computation.** To quantify the smoothness of the spatial domains, we use a nearest-neighbor approach. We extract approximate spatial neighbors for each cell using cuml.NearestNeighbor with 100 neighbors, restricting neighbors to be within the same tissue section. For a given domain set, either from CCF, CellTransformer, or CellCharter, we extract the spatial domain label of the given cell and count the proportion of times that cell is observed in the 100 neighbors. These proportions are averaged across all cells and tissues.

**Classification of domains as discrete or non-discrete.** First, for a given domain at a given resolution (choice of $k$), we average the nearest-neighbor smoothness values over each domain. This process was performed for each compared method and for the CCF domains at 25 domains (matching the CCF division annotations), 354 domains (matching the CCF structure annotations) and 670 domains (matching the CCF substructure annotations). Then, for a given resolution, we used the per-domain average smoothness distribution from the CCF domain sets to derive a per-resolution cutoff. The cutoff was set at the 20th percentile value for each CCF annotation level. We then applied the cutoffs for each resolution to each of the corresponding data-driven domain sets.

To extend the discreteness analysis to domain sets larger than 670, we used the 20th percentile value for the substructure annotations without adjusting them for different values of $k$.

**Linear probing experiments.** We extract neighborhood representations for each of the cells in the Zhuang lab datasets. First, we regress these embeddings on the $(x, y, z)$ coordinates. We then computed the absolute prediction error in terms of the coordinates and then reported the average. We also fit a multi-class logistic regression using the mouse donor identity. For the logistic regression, we use cuml.LogisticRegression with default settings in cuml. For the cell position regression we fit simple least squares using PyTorch via QR decomposition.

**Quantification of spatial contribution to gene expression.** We interpret the accuracy of the gene expression predictions for a given cell as an index of the correlation of an instance of a particular cell type with its surrounding neighbors. To do this, we compute a simple baseline model that predicts average gene expression (computed across the entire Allen Institute for Brain Science mouse dataset) for each cell. We compute the average Pearson correlation for each instance of a given cell type and average across instances to obtain an average Pearson correlation. We then compute a Pearson correlation between each cell's observed gene expression and the Cell-Transformer predictions, averaging similarly across instances of a given cell type. The difference between the baseline and model predictions is displayed, per cell type, and grouped across neurotransmitter types in Supplementary Fig. 12.

**Zhuang lab dataset per-animal CCF comparison.** We contrast two methods of extracting spatial domains from the four animals in the Zhuang lab dataset[12]. We first fix $k$, the number of desired spatial domains. Then we fit one $k$-means model on all of the neighborhood

embeddings for all four (Zhuang 1, 2, 3, and 4) mice together. We also fit a *k*-means model to the embeddings of the mice separately. We then compute the similarity of these region sets using the same method used to quantify differences between CellCharter and CellTransformer by comparing their regional cell type composition vectors.

**Slide-seqV2 analysis.** Initial results with a direct transfer of hyper-parameters to the Langlieb et al.[13] dataset[13] did not produce spatially coherent domains. We therefore implemented two quality control procedures on the raw data. After filtering for coding genes and non-mitochondrial genes, we additionally used only genes that were expressed in >10% of cells in the dataset. At the cellular level, we identified cells with >20% mitochondrial genes and those within the 10th percentile of read depth across each section. We also removed these cells. This left 5019 genes and 4,783,456 cells. We noted that a successful segmentation in the Langlieb et al. dataset required a larger model than the MERFISH ones, using 10 encoder layers rather than 4, which we attributed to the 10X higher number of genes in this dataset versus the 500 in the ABC-WMB Allen 1 dataset. We used a neighborhood size of 50 μm to reduce memory footprint, reasoning the higher cell density in this dataset and higher number of genes would provide enough information for representational richness.

**Other software.** Principal software used in this work includes PyTorch[48], numpy[49], scikit-learn[50], scipy[51], scanpy[52], cuml[53], matplotlib[54], and seaborn[55].

### Reporting summary

Further information on research design is available in the Nature Portfolio Reporting Summary linked to this article.

## Data availability

All MERFISH data used for this publication is available either at the Allen Institute ABC-WMB data portal (https://portal.brain-map.org/atlases-and-data/bkp/abc-atlas; both Allen 1 and Zhuang 1, 2, 3 and 4 datasets), the CZI cellxgene portal (https://cellxgene.cziscience.com/datasets; Zhuang 1, 2, 3, and 4 datasets). Slide-Seq data are available through the BrainCellData website at https://www.braincelldata.org/. Source data are provided with this paper.

## Code availability

Code is publicly available at https://github.com/abbasilab/celltransformer.

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

## Acknowledgements

The authors would like to acknowledge Forrest Collman for help with Neuroglancer data visualization of spatial domains. We acknowledge Patrick Xian for helpful discussions. AJL acknowledges Katharine Z. Yu, Chang Kim, Parker Grosjean, Lee Rao, and Tom Nowakowski for feedback on neuroscientific and machine learning contexts of paper. R.A., B.T., A.J.L., and H.Z. would like to acknowledge support from the Weill Neurohub through the Weill Neurohub's Next Great Ideas Award. R.A. would like to acknowledge support from the National Institute of Mental Health of the National Institutes of Health under award number RF1MH128672 and Sandler Program for Breakthrough Biomedical Research, which is partially funded by the Sandler Foundation. A.J.L. acknowledges support from the UCSF Discovery Fellowship and the UCSF Graduate Research Mentorship Fellowship. Sharing of Allen Institute for Brain Science data through Allen Brain Cell Atlas (and related tools) and registration of the AIBS MERFISH brain to the CCFv3 was funded through 1U24MH130918-01 to L.N.

## Author contributions

A.J.L., H.Z., B.T., and R.A.-A. designed research; A.J.L., and R.A.-A. performed research and analyzed data; A.D., M.K., S.Y., N.L., L.N., H.Z., and B.T. contributed to the data interpretation; A.J.L., and R.A.-A. wrote the paper with contributions from B.T.; All authors edited the manuscript and contributed to the discussions to finalize the manuscript.

## Competing interests

The authors declare no competing interests.
