## [Transparent Peer Review file · Nature Communications]

Data-driven fine-grained region discovery in the mouse brain with transformers

Corresponding Author: Professor Reza Abbasi-Asl

Version 0:

Reviewer comments:

Reviewer #1

(Remarks to the Author)

Summary: This manuscript presents a computational pipeline for the automated detection of spatial domains within tissue characterized with spatial transcriptomic methods. The two-part method first assigns to each cell a high-dimensional “neighborhood” embedding vector, then secondly spatially smooths these embeddings and clusters them to yield the spatial domains. Applied to several spatial transcriptomic datasets, the resulting domains are then thoroughly evaluated and compared to neuroanatomic knowledge. Known neuroanatomy is clearly recapitulated and novel delineations are analyzed for their plausibility, demonstrating how this pipeline is useful as a suggestive and explorative tool for modern neuroanatomy.

Overall evaluation: This is strong manuscript on the axes of its clarity and thoroughness of presentation, on the reasonableness of the computational method, and the evaluation of its implications for neuroanatomy. It provides clear value to the field for thoroughly documenting what regions this class of method identifies in brain tissue.

Where this manuscript could be strengthened is in convincing the reader that this is a better approach than competing methods. For neuroanatomists, this matters because one would like to know which method’s outputs to trust (and where methods agree and disagree). For bioinformaticians, this matters because it is still unclear what the proper way to perform domain detection is for spatial transcriptomics. There were many choices made in the design of this algorithm, and yet there is little in the way of a priori justifications or post hoc evaluations. Clarifying this issue is crucial for cementing the impact of this paper on future work.

Specific comments:

1. Can the specific choices within CellTransformer be justified? For example, why is the output conditioned on cell type? This results in the method clustering the information that is useful for predicting expression but not for predicting cell type compositions. But the alternative is equally reasonable: one could imagine that predicting the cell type distribution as well as the additional information (i.e. just predicting the counts without cell type supplied) would result in a strong signal about neuroanatomy, too. (That cell type conditioning predicts expression better this way, as shown in Supp. Fig. 12, is not relevant for whether it improves domain detection).

2. Why not just cluster on the raw expression? That is, what value does the CellTransformer add? Answering this is crucial for situating this paper in the literature.

Some papers that use this simpler method, which are currently not cited but should be cited in addition to the citations in lines 44-50, are:

<https://bmcbiol.biomedcentral.com/articles/10.1186/s12915-020-00874-5>

<https://www.science.org/doi/10.1126/sciadv.abb3446>

<https://www.biorxiv.org/content/10.1101/2023.06.30.547258v1.full>

What unites these methods (and Shi et al, currently cited) is that they cluster areas directly on the transcriptomic signal, or a rather a linear low-dimensional version of it (PCA or ICA). Why is this simpler approach flawed? Or, how does the current method differ in its predictions than these methods – and which should we trust and why?

(I did appreciate that k-means was run on the cell type count vectors as a baseline in Fig. 2. This is not a replacement for the PCA/ICA comparison, but nevertheless it may be interesting to compare the resulting maps.)

Finally, these missing citations make me question the statements that “To our knowledge, this work provides the first demonstration that large scale data-driven discovery of domains at CCF-like resolution can be based on spatial transcriptomics data.” in lines 77-79 as well as repeated in lines 580-581 in the discussion. Removing the claim to primacy, these sentences could read simply “This work demonstrates that large scale...”

3. The current method spatially smooths the embeddings before clustering. This of course biases the resulting clustering as intended; the neighbor graph is changed so that cells that are spatial neighbors are more likely to be neighbors in embedding space, biasing the clustering. This raises a set of questions:

- This is an artificial adjustment which overrides signals in the data and especially obscures signals in the data that are spatially non-contiguous. These may very well be ‘real’ and are at least interesting. As is nicely explored in Supp. Fig. 11, such organization appears to exist in the striatum for example.

- A second issue is that of comparison. The other methods compared (SPIRAL, etc.) were presented without this spatial smoothing step applied. This does not seem like a fair comparison. In Supplementary Figure 7, for example, it is pointed out that CellCharter “loses spatial coherence”, which makes a less desirable method. But on these particular slides, how much is the spatial coherence of CellTransformer due to the smoothing step? As another example, in Figure 2c, CellTransformer is the best-performing model in the category of ‘neighbor spatial similarity’. This measure will be affected by smoothing; how much is smoothing a factor here? Presenting the current step without spatial smoothing in these instances would allow a fair comparison.

Overall, this was a pleasure to read. The constant dialogue between neuroanatomical knowledge and algorithmic analysis is unusually strong. Items I particularly enjoyed were

- Line 178, confirmation of granular L4 in motor cortex
- 221, literature comparison in caudoputamen
- Fig 2b, the highlighting of motor and sensory layers
- Analysis of consistency across datasets

Yet it is crucial to discuss and evaluate why readers should trust these results as opposed to other similar clustering approaches – and to discuss how these seemingly reasonable choices affect the outcome of domain detection.

There are many ways of addressing these concerns, from adding justifying text to adding more thorough visualizations of comparisons and with more methods, and potentially by adding new quantitative metrics.

Why the current evaluations are insufficient:

- Throughout the manuscript the spatial distribution of cell types is used as corroborating evidence of area boundaries. While this is nice to see, it is not conclusive as cell type composition is not the ground truth way to delineate areas. (If it were, the k-means on cell type compositions run here as a baseline would be the preferred method. Using cell type composition to predict brain areas via supervised machine learning does not work as well as pseudobulk genes; Chen et al. 2024 in Nature presents one such analysis). Furthermore, as a distance metric (Fig 2d,e) it is strongly sensitive to some changes and insensitive to others. (To the question of “are these areas similar in both cartographies?”, the straightforward method would be to ask the spatial overlap between matching areas. There are many ways of doing this, but the key thing is that the spatial information is the domain of comparison. Could this be done, or was there a reason this is not possible?)
- Spatial smoothing not evenly applied across methods
- No discussion of model design choices, other than k

Minor comments and suggestions (not mandatory):

1. It would be nice to see a discussion of continuous gradients vs. discrete areas. By clustering CellTransformer representations, this manuscript embodies the assumption that the brain is best conceptualized as a tapestry of discrete areas, presumably with discrete functions. However, nature may well compose with gradients. Indeed, the present manuscript notes opposing gradients of single genes in the subiculum in Figure 3, and there is a wealth of papers that describe similar genetic gradients across the brain, both in cortical depth and in rostral/caudal and medial/lateral axes. Furthermore, the existence of gradients is arguably supported by this paper’s finding that there is no finite number k of regional clusters that maximizes a stability criterion, suggesting (by this criterion) an infinite number of regions. The discussion should note the possibility of such organizations.

If the authors are inclined, it could be interesting to plot out the CellTransformer embeddings directly without clustering them. (The top N PCs, at least). These could be examined in detail in areas like the subiculum.

2. In the CCF cortical areas are defined by function primarily and by layer secondarily. However, the transcriptomic differences due to layer are generally much larger than the transcriptomic differences across cortical columns. This suggests that transcription alone may not be a sufficient signal to delineate brain areas – so long as it is desired that brain areas have distinct ‘functions’ as is often defined by physiologists. I recognize this debate is a bit out of scope of this manuscript.

However, it would be nice to note in the discussion the tension between function-defined and transcription-defined notions of cortical areas.

3. In Lines 559-560, it is noted that for Slide-SeqV2 data it was necessary to use a 10-layer encoder instead of a 4 layer encoder in order to maintain spatial coherence. If these diagnostic plots still exist, it would be interesting to include them as supplementary information as examples of what can go wrong. Practitioners using this method on new data modalities would be helped along by this discussion of common failure modes.

(Remarks on code availability)

Reviewer #2

(Remarks to the Author)

The authors present CellTransformer, a method for clustering cells in spatial transcriptomic data. The approach first trains a model to predict a single cell's gene expression based on the subclasses and expression profiles of neighboring cells. The resulting latent representations are then used for clustering into domains.

While the authors visually compare their results to the CCF atlas, they do not quantify this alignment using standard metrics such as the Adjusted Rand Index (ARI) or Normalized Mutual Information (NMI). Instead, they rely on indirect measures, such as the correlation of cell type compositions (Figure 2f), which provide limited insight into clustering accuracy.

Although a smoothness metric is provided, a basic evaluation of spatial coherence is missing: what proportion of domains are spatially contiguous? Figure S11 (for $k=1300$) suggests that many clusters lack contiguity, and the method does not appear to enforce this.

The inclusion of spatial information into the transformer model appears suboptimal. The use of a square neighborhood - rather than a more biologically intuitive circular or spherical region and the omission of distance from the masked cell as an input feature (position encoding). These design choices may limit the model's ability to capture spatial relationships effectively.

A UMAP projection of the latent space would help clarify whether discrete clusters emerge in this representation. If Figure 1b depicts such a projection, it suggests that cluster boundaries are not clear.

The comparison to CellCharter is unfair, as CellCharter does not utilize subclass information as input. The 338 subclasses provided to CellTransformer, derived from an external, transcriptome-wide dataset, effectively introduce a genome-wide imputation step supported by substantial prior data. An ablation study removing this subclass information is needed to assess whether performance drops to CellCharter's level, ensuring a fair benchmark.

The cross-animal analyses in Figure 6 are a valuable addition, though their interpretation is somewhat circular since clustering relies on combined embeddings across slices. Notably, Figure 6b indicates that 1 in 5 clusters are inconsistent across three animals for $k > 650$, casting doubt on the choice of $k=1300$ throughout the manuscript. Justification for this parameter selection is lacking.

Minor

-The term 'cell types' is applied to the cluster level data ('5274 cell types' in Figure 2g for example).

-No classifier loss curve is provided. Does the loss plateau, or is more data needed to predict the expression of the focus cell?

(Remarks on code availability)

Reviewer #3

(Remarks to the Author)

(Remarks on code availability)

Version 1:

Reviewer comments:

Reviewer #1

(Remarks to the Author)

Thank you for the thorough responses to our questions. All our concerns have been addressed and this will be a nice addition to an exciting area.

Looking at the supplementary plots, it is clear that the method with the original set of design choices does indeed outperform both the ablated model without cell type conditioning and the comparison to clustering directly on the MERFISH signal. The addition of the new metric of directly quantifying the overlap with CCF further strengthens the comparison. More importantly, though, the inclusion of these alternative design choices will be helpful to practitioners extending this method in the future. Most importantly, we thank the authors for including the non-smoothed CellTransformer in metrics and visualizing the differences. It is now clear how much of the shown effects are due to this step. It appears to have only affected the results only modestly

(Remarks on code availability)

Reviewer #2

(Remarks to the Author)

The authors have reasonably addressed my points.

(Remarks on code availability)

Reviewer #3

(Remarks to the Author)

(Remarks on code availability)

The repository appears complete and well organized, with instructions to reproduce figures and download the data. I did not attempt to reproduce figures myself.

Response to the reviewers' comments

We thank the reviewers for their constructive feedback and insightful comments. Below, we provide a point-by-point response (in blue) to each of the referee's comments.

REVIEWER COMMENTS

Reviewer #1 (Remarks to the Author):

Summary: This manuscript presents a computational pipeline for the automated detection of spatial domains within tissue characterized with spatial transcriptomic methods. The two-part method first assigns to each cell a high-dimensional "neighborhood" embedding vector, then secondly spatially smooths these embeddings and clusters them to yield the spatial domains. Applied to several spatial transcriptomic datasets, the resulting domains are then thoroughly evaluated and compared to neuroanatomic knowledge. Known neuroanatomy is clearly recapitulated and novel delineations are analyzed for their plausibility, demonstrating how this pipeline is useful as a suggestive and explorative tool for modern neuroanatomy.

Overall evaluation: This is strong manuscript on the axes of its clarity and thoroughness of presentation, on the reasonableness of the computational method, and the evaluation of its implications for neuroanatomy. It provides clear value to the field for thoroughly documenting what regions this class of method identifies in brain tissue.

Where this manuscript could be strengthened is in convincing the reader that this is a better approach than competing methods. For neuroanatomists, this matters because one would like to know which method's outputs to trust (and where methods agree and disagree). For bioinformaticians, this matters because it is still unclear what the proper way to perform domain detection is for spatial transcriptomics. There were many choices made in the design of this algorithm, and yet there is little in the way of a priori justifications or post hoc evaluations. Clarifying this issue is crucial for cementing the impact of this paper on future work.

Specific comments:

1. Can the specific choices within CellTransformer be justified? For example, why is the output conditioned on cell type? This results in the method clustering the information that is useful for predicting expression but not for predicting cell type

compositions. But the alternative is equally reasonable: one could imagine that predicting the cell type distribution as well as the additional information (i.e. just predicting the counts without cell type supplied) would result in a strong signal about neuroanatomy, too. (That cell type conditioning predicts expression better this way, as shown in Supp. Fig. 12, is not relevant for whether it improves domain detection).

We thank the reviewer for the comments and clear and constructive feedback. The reviewer raises an important point that alternative model constructions may also be performant, and a more detailed exploration would help convince potential readers about the benefits of our proposed class of models. Based on these comments, we trained two additional variants of CellTransformer on the AIBS 1 dataset [1]. One of these models did not use cell type conditioning in the decoder portion of the network, and the other did not use any cell type information in either the encoder or decoder portions of the network. We then evaluated both models by computing the similarity between identified regions and CCF, as well as the neighborhood smoothness score (similar to other evaluations in the manuscript). Additionally, we included metrics for regions identified from both smoothed and non-smoothed embeddings. These variants perform competitively with the base CellTransformer version. The base model domains were 3.0% more similar to CCF than the model without cell type decoding, and 5.4% more similar than the model without cell type in either the encoder or decoder (See the newly produced **Supplementary Figure 9a**, also included below). The smoothed and unsmoothed CellTransformer domains were also very similarly spatially smooth (**Supplementary Figure 9b**), even when the number of domains was extended to 2,000. CellTransformer variants also produced relatively little decrease in spatial smoothness at greater than 1,000 domains, whereas CellCharter smoothness sharply declined. Similarly to the CCF comparison, the model without cell type decoding performed better than the model without cell type in both the encoder and decoder. We also computed normalized mutual information between the CellTransformer base model and the different variants (smoothing, without cell type information; **Supplementary Figure 9c**), finding that all variants were similar to the base model. Taken together, these results indicate that cell type conditioning is an important component of the model that increases similarity to CCF and spatial uniformity. This may be because the cell types, which were fit using whole-transcriptome scRNA-seq profiling, implicitly comprise a whole-transcriptome imputation step. However, we also note that the CellTransformer variant without any cell type information still performs better than CellCharter with respect to similarity to CCF and spatial coherence.

We have produced the following supplementary figure and included this figure in the revised manuscript:

Supplementary Figure 9. Comparison of different CellTransformer variants (smoothed versus unsmoothed embeddings; models with and without cell type information in the transformer encoder and decoder) in the Allen 1 dataset. (a.) Average max correlation of data-driven domains to CCF. (b.) Neighborhood spatial similarity, averaged all cells by models.

“CellTransformer utilizes cell type information in both the encoder and decoder portions of the network. To understand the impact of this choice, we trained two additional variants of CellTransformer, one without cell type information in the decoder and one without cell type information in the encoder and the decoder (only expression). We generated spatial domains from these models’ embeddings with smoothing, performed identically as in the base model. These variants perform competitively with the base CellTransformer version. The base model domains were 3.0% more similar to CCF than the model without cell type decoding, and 5.4% more similar than the model without cell type in either the encoder or decoder (**Supplementary Figure 9a**). The smoothed and unsmoothed CellTransformer domains were also very similarly spatially smooth (**Supplementary Figure 9b**), even when the number of domains was extended to 2,000. CellTransformer variants also produced relatively little decrease in spatial smoothness at greater than 1,000 domains, whereas CellCharter smoothness sharply declined. Similarly to the CCF comparison, the model without cell type decoding performed better than the model without cell type in both the encoder and decoder. We also computed normalized mutual information between the CellTransformer base model and the different variants (smoothing, without cell type information; **Supplementary Figure 9c**), finding that all variants were similar to the base model. Taken together, these results indicate that cell type conditioning is an important component of the model that increases similarity to CCF and spatial uniformity. This may be because the cell types, which were fit using whole-transcriptome scRNA-seq profiling, implicitly comprise a whole-transcriptome imputation step. However, we also note that the CellTransformer variant without any cell type information still performs better than CellCharter with respect to similarity to CCF and spatial coherence. “

It is worthwhile to mention that we designed our original model to condition on output cell type for several reasons. First, our architecture is heavily inspired by the masked autoencoder [2] / masked language modeling [3] framework, which both utilize conditioning information in the form of positional embeddings. We also took inspiration from the node-centric expression modeling approach adopted in Fischer et al. (2023, Nature Biotechnology) [4], which uses a similar architecture without pooling and graph attention, and also conditions expression predictions on cell type. We aimed to employ a similar method as in Fischer et al. (2023, Nature Biotechnology) to quantify cell-cell interactions in future work, especially considering our analyses in the Supplementary

Fig. 12 appear to identify biologically plausible differences in niche composition contributors to gene expression. We have added the following text to the Results section to clarify the reasoning behind this decision:

“...Importantly, during this process, only the mask token and the neighborhood representation can attend to each other. This operation captures a hierarchical encoding and decoding process where low-level information (gene and cell type) is produced at the cell token level and aggregated into a high-level representation. This high-level representation is then used to conduct the reverse decoding process (prediction of gene expression from cell type and tissue context information). This construction strongly resembles that of masked language models [2], masked autoencoders [3], where masked predictions are generated based on a conditioning signal (position encodings). In our model, position-based conditioning is replaced by cell type conditioning, similarly to the NCEM [4] model. However, unlike NCEM[4], we aggregate information across tokens (nodes) in a cellular neighborhood using a learned pooling which strongly bottlenecks the information distributed across the tokens prior to masked cell prediction...”

[1] Yao, Z., van Velthoven, C. T. J., Kunst, M., Zhang M., McMillen, D., Lee, C., Jung, W., et al. Nature (2023).

[2] He. K., Chen, X., Xie, S., Li, Y., Dollár, P., Girschick, R. Masked Autoencoders are Scalable Vision Learners. arXiv (2021).

[3] Devlin, J., Ming-Wei, C., Lee, K., Toutanova, K. BERT: Pre-training of deep bidirectional transformers for language understanding. arXiv (2019).

[4] Fischer, D. S., Schaar, A. C., Theis, F. J. Modeling intercellular communication in tissues using spatial graphs of cells. Nature Biotechnology (2023).

2. Why not just cluster on the raw expression? That is, what value does the CellTransformer add? Answering this is crucial for situating this paper in the literature.

Some papers that use this simpler method, which are currently not cited but should be cited in addition to the citations in lines 44-50, are:

<https://bmcbiol.biomedcentral.com/articles/10.1186/s12915-020-00874-5>

<https://www.science.org/doi/10.1126/sciadv.abb3446>

<https://www.biorxiv.org/content/10.1101/2023.06.30.547258v1.full>

What unites these methods (and Shi et al, currently cited) is that they cluster areas directly on the transcriptomic signal, or a rather a linear low-dimensional version of it (PCA or ICA). Why is this simpler approach flawed? Or, how does the current method differ in its predictions than these methods – and which should we trust and why?

(I did appreciate that k-means was run on the cell type count vectors as a baseline in Fig. 2. This is not a replacement for the PCA/ICA comparison, but nevertheless it may be interesting to compare the resulting maps.)

Finally, these missing citations make me question the statements that “To our knowledge, this work provides the first demonstration that large scale data-driven discovery of domains at CCF-like resolution can be based on spatial transcriptomics data.” in lines 77-79 as well as repeated in lines 580-581 in the discussion. Removing the claim to primacy, these sentences could read simply “This work demonstrates that large scale...”

We agree that gene expression-based domain detection represents an important line of work in the spatial transcriptomics analysis field. To directly demonstrate the benefits of our approach over direct clustering of gene expression, we used *k*-means clustering to cluster single cell MERFISH gene expression profiles with *k*=25, 354, and 670 clusters. We produced an amended version of Figure 2 which includes this baseline (two subpanels of Figure 2 shown below, with this gene expression baseline shown in green). Clustering on gene expression produces much less spatially coherent domains than CellTransformer and similarly inspired deep learning models. This representation is also much less performant than our previously-generated baseline using cell type count vectors. Overall smoothness is low (**Figure 2c**), with CellTransformer (smoothed) domains 419% smoother than the gene expression baseline. Gene expression clustering also produced much less similar domains to CCF than other methods, producing the lowest overall similarity amongst methods tested when using 1:1 domain matching as in **Figure 2f**.

Figure 2. Representative images of spatial domains discovered using CellTransformer on the Allen 1 dataset (53 coronal sections and 500 gene MERFISH panel) and comparison to CCF... (c.) Spatial homogeneity (see **Methods**) of domains from different methods including recently published methods CellCharter and SPIRAL... (f.) Average

Pearson correlation (averaging over number of domains and method) of optimal matched pairs between data-driven and CCF regions, where CCF regions are only allowed to pair with one data-driven region per comparison. Matches fit using linear programming.

We thank the reviewers for pointing out papers by Partel et al., (BMC Biology), Ortiz et al. (Science Advances), and Maher et al. (biorXiv). We have included these references in the text of the manuscript. We posit that, as reviewer #2 noted, inclusion of cell type information acts as an implicit whole-genome imputation. As we show in Supplementary Figure 9 (displayed above), this improves performance and allows for the extraction of domains that are more concordant with prior human knowledge and are more spatially coherent. We have also revised the language related to the claim to primacy in the revision, and we highlight that our study is one of the first to introduce a highly scalable method with the ability to discover a significantly larger number of domains compared to existing models.

3. The current method spatially smooths the embeddings before clustering. This of course biases the resulting clustering as intended; the neighbor graph is changed so that cells that are spatial neighbors are more likely to be neighbors in embedding space, biasing the clustering. This raises a set of questions:

- This is an artificial adjustment which overrides signals in the data and especially obscures signals in the data that are spatially non-contiguous. These may very well be 'real' and are at least interesting. As is nicely explored in Supp. Fig. 11, such organization appears to exist in the striatum for example.
- A second issue is that of comparison. The other methods compared (SPIRAL, etc.) were presented without this spatial smoothing step applied. This does not seem like a fair comparison. In Supplementary Figure 7, for example, it is pointed out that CellCharter "loses spatial coherence", which makes a less desirable method. But on these particular slides, how much is the spatial coherence of CellTransformer due to the smoothing step? As another example, in Figure 2c, CellTransformer is the best-performing model in the category of 'neighbor spatial similarity'. This measure will be affected by smoothing; how much is smoothing a factor here? Presenting the current step without spatial smoothing in these instances would allow a fair comparison.

We thank the reviewers for this comment. We have added a series of figures to the manuscript comparing the smoothed with non-smoothed CellTransformer embeddings. The principal reason for smoothing was to produce spatially contiguous and connected domains in concordance with neuroanatomical

conventions, but we agree that such non-contiguous domains may represent actual biology. As shown in the above **Supplementary Figure 9**, whether or not smoothing is applied, CellTransformer performs much better than comparator methods with respect to spatial smoothness and similarity to CCF. We also point out an additional detail to support the decision to smooth the embeddings. In response to Reviewer 2's feedback, we developed a metric quantifying the proportion of discrete domains produced by a given method (see page 16 of response; Supplementary Figure 10 and Figure 2d in revised manuscript). This analysis shows that smoothing the domains maintains a high number of spatially discrete domains, while also making the distribution of region-wise smoothness more similar to CCF.

To provide additional visual comparison between the smoothed with non-smoothed CellTransformer, we have now added the **Supplementary Figure 11**, visualizing regions both with and without smoothing. Specifically, we note that in sections more anterior to the striatum, no heterogeneity can be observed (sections 55 and forward), and in sections following (section 43; more sections were omitted due to space limitations), the non-contiguous domains are not observed. This can be confirmed in **Supplementary Figure 12**, where we provide a larger series of domains that demonstrate that non-contiguous domains are limited to roughly sections 45 to 52 and only in the striatum.

We have added the following text to the Supplementary Note 1 to contextualize this:

"**Supplementary Figure 11**, which shows roughly the same sections as **Supplementary Figure 18** (containing the striatal non-contiguous domains), also shows that in the unsmoothed embeddings, non-contiguous domains are limited to the striatum and only between sections 45 and 52. More anterior sections (past section 55) do not demonstrate any non-contiguous domains and prior to section 43 there are also no non-contiguous domains. This can be seen at a larger scale in **Supplementary Figure 12** across the entire Allen 1 dataset. "

Supplementary Figure 11. Comparison of spatial domains discovered at $k=1300$ domains when using smoothed versus unsmoothed CellTransformer embeddings and focusing on the isolated case of non-uniform spatial domains discussed in **Supplementary Note 1**. Although overall similarity is high, some lamina, particularly in the cortex (sections 45, 43) display slightly altered boundaries; a subset of these are highlighted with black arrows. All sections are from the ABC-MWB (Allen 1) dataset.

Supplementary Figure 12. CellTransformer spatial domains (left) and the corresponding CCF annotations (right) organized in 3 columns for roughly half of the sections in the Allen 1 dataset, approximately every other section. CellTransformer domains were calculated at $k=1300$ clusters without smoothing.

Overall, this was a pleasure to read. The constant dialogue between neuroanatomical knowledge and algorithmic analysis is unusually strong. Items I particularly enjoyed were

- Line 178, confirmation of granular L4 in motor cortex
- 221, literature comparison in caudoputamen
- Fig 2b, the highlighting of motor and sensory layers
- Analysis of consistency across datasets

Yet it is crucial to discuss and evaluate why readers should trust these results as opposed to other similar clustering approaches – and to discuss how these seemingly reasonable choices affect the outcome of domain detection.

There are many ways of addressing these concerns, from adding justifying text to adding more thorough visualizations of comparisons and with more methods, and potentially by adding new quantitative metrics.

Why the current evaluations are insufficient:

- Throughout the manuscript the spatial distribution of cell types is used as corroborating evidence of area boundaries. While this is nice to see, it is not conclusive as cell type composition is not the ground truth way to delineate areas. (If it were, the k-means on cell type compositions run here as a baseline would be the preferred method. Using cell type composition to predict brain areas via supervised machine learning does not work as well as pseudobulk genes; Chen et al. 2024 in Nature presents one such analysis). Furthermore, as a distance metric (Fig 2d,e) it is strongly sensitive to some changes and insensitive to others. (To the question of “are these areas similar in both cartographies?”, the straightforward method would be to ask the spatial overlap between matching areas. There are many ways of doing this, but the key thing is that the spatial information is the domain of comparison. Could this be done, or was there a reason this is not possible?)
- Spatial smoothing not evenly applied across methods
- No discussion of model design choices, other than k

We thank the reviewers for this clear feedback and commentary. In response to this, we quantified direct spatial overlap between areas in CCF and data-driven ones by computing the normalized mutual information (NMI) and adjusted Rand Index (ARI). NMI directly quantifies similarity between two clusterings. ARI quantifies pairs of data points that are in the same clusters across two label sets and adjusts for expected similarity between random data points (**Supplementary Figure 15**). CellTransformer performs better than other methods across all resolutions for both metrics. For NMI, CellTransformer (both with and without embedding smoothing) performs 13% better

than CellCharter, the closest comparator method. For ARI, CellTransformer (with smoothing) performs 24% better than CellCharter and without smoothing performs 22% better than CellCharter. We note that registering MERFISH datasets to CCF is challenging due to the difficulty of comparing a flat, thin tissue section to a dense voxel image, which may explain overall low agreement as assessed by ARI.

Supplementary Figure 15. Direct quantification of overlap between CCF and data-driven domains in the Allen 1 dataset. (a.) Normalized mutual information comparison of different models data-driven domains versus CCF. (b.) Adjusted Rand Index comparison of different models data-driven domains versus CCF.

Minor comments and suggestions (not mandatory):

1. It would be nice to see a discussion of continuous gradients vs. discrete areas. By clustering CellTransformer representations, this manuscript embodies the assumption that the brain is best conceptualized as a tapestry of discrete areas, presumably with discrete functions. However, nature may well compose with gradients. Indeed, the present manuscript notes opposing gradients of single genes in the subiculum in Figure 3, and there is a wealth of papers that describe similar genetic gradients across the brain, both in cortical depth and in rostral/caudal and medial/lateral axes. Furthermore, the existence of gradients is arguably supported by this paper's finding that there is no finite number k of regional clusters that maximizes a stability criterion, suggesting (by this criterion) an infinite number of regions. The discussion should note the possibility of such organizations.

The reviewers raise an important point. To address this, we added the following text to the discussion:

"We note that our objective is not to suggest that the domains discovered by our method are a definitive, normative set of brain regions, and we did not conduct any further validation studies to support these claims. Nor is it to strongly assert that the brain is composed of discrete brain regions, as opposed to a composition of gene expression gradients. Our objective was to develop a tool that would operationally

identify plausible brain structures in a data-driven way and to satisfy a neuroanatomical convention for discrete domains. To support this goal, we used *k*-means clustering due to its capability to be mini-batched during fitting and because it produced biologically plausible domains; a probabilistic framework such as Gaussian mixture modeling (as used by CellCharter) may also have worked well, and possibly would have produced similarly plausible domains. Indeed, the fact that no finite number of domains was identified via our stability criterion suggests the possibility of a significantly larger number of domains.“

If the authors are inclined, it could be interesting to plot out the CellTransformer embeddings directly without clustering them. (The top N PCs, at least). These could be examined in detail in areas like the subiculum.

We thank the reviewers for their suggestion and agree that this would be an interesting addition. We produced a visualization of the first three principal components of the CellTransformer embeddings (together 19.5% of dataset-wise variance in the Allen 1 dataset). We visualized the same sections as in the previous draft, Supplementary Figure 11 (containing non-contiguous domains in striatum), in **Supplementary Figure 2**. We have now included this figure in the supplement.

Supplementary Figure 2. Visualizations of leading principal components of CellTransformer embeddings across several tissue sections in the Allen 1 dataset¹. (a-c.) Principal components 1-3, in total constituting 19.5% of dataset-wide variance, across five sequential sections. (d.) Scree plot of variance explained by component.

2. In the CCF cortical areas are defined by function primarily and by layer secondarily. However, the transcriptomic differences due to layer are generally much larger than the transcriptomic differences across cortical columns. This suggests that transcription alone may not be a sufficient signal to delineate brain areas – so long as it is desired that brain areas have distinct ‘functions’ as is often defined by physiologists. I recognize this debate is a bit out of scope of this manuscript. However, it would be nice to note in the discussion the tension between function-defined and transcription-defined notions of cortical areas.

We have now added the following sentences to the Discussion section:

“While CellTransformer was able to reveal patterns aligned with functionally defined cortical regions, we recognize that functional distinctions, as defined by physiological measurements, may not always map neatly onto transcriptional boundaries. Including additional data modalities—such as electrophysiological recordings—could further refine and validate transcriptomic parcellations. This underscores the value of integrative frameworks that bring together transcriptomic, anatomical, and physiological data to better understand cortical organization.”

3. In Lines 559-560, it is noted that for Slide-SeqV2 data it was necessary to use a 10-layer encoder instead of a 4 layer encoder in order to maintain spatial coherence. If these diagnostic plots still exist, it would be interesting to include them as supplementary information as examples of what can go wrong. Practitioners using this method on new data modalities would be helped along by this discussion of common failure modes.

We thank the reviewer for the comment. We specifically designed a larger encoder with 10 layers for this data to allow the model to handle the much larger number of cells in the dataset compared to MERFISH data. The goal here was to showcase the utility of CellTransformer on a new spatial transcriptomics modality, and the panel e in Figure 6 demonstrates that our model is able to identify regions in the Slide-SeqV2 data. We have now clarified the rationale for this choice in the text and noted that future work focused on Slide-Seq data should include hyperparameter tuning analysis.

Reviewer #2 (Remarks to the Author):

The authors present CellTransformer, a method for clustering cells in spatial transcriptomic data. The approach first trains a model to predict a single cell’s gene expression based on the subclasses and expression profiles of neighboring cells. The resulting latent representations are then used for clustering into domains.

While the authors visually compare their results to the CCF atlas, they do not quantify this alignment using standard metrics such as the Adjusted Rand Index (ARI) or Normalized Mutual Information (NMI). Instead, they rely on indirect measures, such as the correlation of cell type compositions (Figure 2f), which provide limited insight into clustering accuracy.

We thank the reviewer for this feedback. To address this, we computed the NMI and ARI for CellTransformer domains versus other methods, compared to CCF. CellTransformer performs better than other methods across all resolutions for both metrics. For NMI, CellTransformer (both with and without embedding smoothing) performs 13% better than CellCharter, the closest comparator method. For ARI, CellTransformer (with smoothing) performs 24% better than CellCharter, and without smoothing, performs 22% better than CellCharter. We note that registering MERFISH datasets to CCF is challenging due to the difficulty of comparing a flat, thin tissue section to a dense voxel image, which may explain the overall low agreement as assessed by ARI.

Supplementary Figure 15. Direct quantification of overlap between CCF and data-driven domains in the Allen 1 dataset. (a.) Normalized mutual information comparison of different models data-driven domains versus CCF. (b.) Adjusted Rand Index comparison of different models data-driven domains versus CCF.

Although a smoothness metric is provided, a basic evaluation of spatial coherence is missing: what proportion of domains are spatially contiguous? Figure S11 (for $k=1300$) suggests that many clusters lack contiguity, and the method does not appear to enforce this.

We thank the reviewer for this constructive comment. To address this, we have performed an analysis quantifying the proportion of spatially contiguous domains at different numbers of domains. We have added the following subpanel to Figure 2, as well as the following text to the manuscript:

Figure 2. Representative images of spatial domains discovered using CellTransformer on the Allen 1 dataset (53 coronal sections and 500 gene MERFISH panel¹) and comparison to CCF...**(d.)** Proportion of discrete domains by model as compared to CCF at the same resolution. Single-cell level spatial smoothness was averaged over each domain instead of averaged over the entire dataset as in **(c.)**. For each CCF annotation level, we computed a threshold as the 20th percentile of per-domain spatial smoothness values. We applied this threshold to the distribution of data-driven domains at the same resolution...

“...For a given CCF annotation resolution, we then computed a threshold based on the 20th percentile of per-domain averaged CCF smoothness values. This adaptive metric allows us to compare fairly to human annotations at different resolutions. CellTransformer domains fit with and without smoothing both perform well, with CellCharter, SPIRAL, and the gene expression baselines decreasing significantly at higher domain numbers (**Figure 2d**). We also computed the proportion of spatially discrete domains for resolutions from 700 to 2,000 data-driven domains using the 20th percentile cutoff from the k=670 CCF annotation level, demonstrating the superior performance of CellTransformer versus CellCharter, where the proportion of discrete domains significantly diminishes (**Supplementary Figure 10c**), unlike CellTransformer. The unsmoothed CellTransformer embedding workflow was most performant; we reasoned that the isotropic Gaussian smoothing employed may have eroded fine laminar boundaries, despite removing the isolated non-uniform domains in the striatum (**Supplementary Note 1**).”

This shows that using a per-resolution threshold based on CCF, CellTransformer produces much more spatially discrete domains than other methods.

We also include the following supplementary figure, quantifying the distribution of per-domain average smoothness of CellTransformer and CellCharter domains, compared with CCF, at k=670. This shows that at the regional level, the smoothness characteristics of CellTransformer and hand-drawn annotations in CCF are broadly similar. Additionally, we show that extrapolating a discreteness cutoff fit at k=670 hand-drawn CCF regions to larger numbers (from 700 to 2000 domains), CellTransformer domains are much more performant than CellCharter, the next best model.

Supplementary Figure 10. Visualizations of the distribution of per-domain spatial smoothness comparing CCF and various data-driven spatial domains. All sections are from the ABC-MWB (Allen 1) dataset. **(a.)** Distribution of per-region spatial smoothness for 670 CCF domains compared with 670 CellCharter and 670 CellTransformer domains (shown with smoothing). **(b.)** Distribution of per-region spatial smoothness for 670 CCF domains, compared with CellCharter and CellTransformer at 1300 domains. **(c.)** Proportion of discrete domains using a fixed cutoff at 20th percentile of per-region CCF smoothness values (0.381), applied to CellCharter and CellTransformer from 700 to 2,000 domains.

The inclusion of spatial information into the transformer model appears suboptimal. The use of a square neighborhood - rather than a more biologically intuitive circular or spherical region and the omission of distance from the masked cell as an input feature (position encoding). These design choices may limit the model's ability to capture spatial relationships effectively.

We thank the reviewer for their comment. While the choice of a circular neighborhood could certainly be used, we decided on a square neighborhood for two reasons: 1. Our future applications will include datasets that are pixel-wise voxel-wise and a square window is a better fit for these applications, and 2. as the lookup is 10.4% faster than a circular one and increases overall computation time. We have now updated our GitHub codebase and included the implementation of a circular neighborhood choice for the users.

A UMAP projection of the latent space would help clarify whether discrete clusters emerge in this representation. If Figure 1b depicts such a projection, it suggests that cluster boundaries are not clear.

We apologize for the miscommunication. Figure 1b does not depict a UMAP of the latent space and is merely for illustration. We have clarified this in this amended caption for Figure 1:

“Note that the latent variable point cloud shown is merely for illustration and does not correspond to the actual CellTransformer latent space.”

We generated a UMAP of the CellTransformer embedding space and have added this to the manuscript. However, we caution against using the UMAP to interpret the relative separability of the clusters. We note that UMAP’s propensity to distort distances between data points (Chari et al. [1] and Lause et al. [2]) has been studied extensively. Both Chari et al. and Lause et al. note the importance of following up with targeted studies, as we have throughout the manuscript, to understand the behavior of high-dimensional data. We note that it is difficult for any 2D dimensionality reduction method to capture variability and inter-point distances in a high-dimensional space in a coherent and accurate manner. In particular, it is difficult for any dimensionality reduction algorithm to capture the fact that spatial domains which are close together in space (either within a section or across sections) will necessarily also be close to a larger number of other domains due to the 3D geometry of the brain and its molecular autocorrelation.

Supplementary Figure 7. UMAP of smoothed CellTransformer embeddings on the Allen 11 dataset. **(a.)** UMAP colored by CCF division (25 total) annotation for each cell; the “fiber-tracts-unassigned”, “mfbs”, “cbf”, “eps”, “cm”, “V4”, “VL”, “AQ”, “scwm”, and “V3” areas were collapsed into the “Fiber tracts and white matter” label to make the visualization simpler. **(b.)** UMAP colored by CellTransformer ($k=25$) domains. Domains are grouped by the CCF division with the greatest spatial overlap. Domains corresponding to the striatum (domains 1 and 16) are highlighted with black arrows. Abbreviations: mfbs-medial forebrain bundle system; cbf-cerebellum related fiber tracts; cm-cranial nerves; eps-external plexiform layer; V3-third ventricle; V4-fourth ventricle; VL-lateral ventricle; scwm-superior colliculus commissure; AQ-cerebral aqueduct; MY-medulla; P-pons; MB-midbrain; HPF-hippocampal formation; OLF-olfactory bulb; TH-thalamus; CTXsp-cortical subplate; HY-hypothalamus; STR-striatum; PAL-pallidum.

- [1] Chari, T., Pachter, L. The specious art of single-cell genomics. PLOS Computational Biology, 2023.
- [2] Lause, J., Berens, P., Kobak, D. The art of seeing the elephant in the room: 2D embeddings of single-cell data do make sense. PLOS Computational Biology, 2024.

The comparison to CellCharter is unfair, as CellCharter does not utilize subclass information as input. The 338 subclasses provided to CellTransformer, derived from an external, transcriptome-wide dataset, effectively introduce a genome-wide imputation step supported by substantial prior data. An ablation study removing this subclass information is needed to assess whether performance drops to CellCharter’s level, ensuring a fair benchmark.

We thank the reviewer for this comment. One of the benefits of our approach is to take advantage of cell type information in the model, which improves performance. To support this claim in the manuscript, we conducted the following analysis. An additional model was trained on the Allen 1 dataset without any cell type information at all. We then benchmarked similarity to human annotations via CCF and neighborhood spatial similarity. As shown in the **Supplementary Figure 9 (also included below)**, CellTransformer without any cell type information at all (purple line) performs worse than the base CellTransformer but performs significantly better than the CellCharter.

Supplementary Figure 9. Comparison of different CellTransformer variants (smoothed versus unsmoothed embeddings; models with and without cell type information in the transformer encoder and decoder) in the Allen 1 dataset¹. (a.) Average max correlation of data-driven domains to CCF. (b.) Neighborhood spatial similarity, averaged all cells by models. (c.) Normalized mutual information comparing the base CellTransformer model domains with variants.

The cross-animal analyses in Figure 6 are a valuable addition, though their interpretation is somewhat circular since clustering relies on combined embeddings

across slices. Notably, Figure 6b indicates that 1 in 5 clusters are inconsistent across three animals for $k > 650$, casting doubt on the choice of $k=1300$ throughout the manuscript. Justification for this parameter selection is lacking.

We thank the reviewer for their comment. The Zhuang dataset is composed of sections from four animals. One animal (“Zhuang 1”) has 147 sections. The next (“Zhuang 2”) has 66 sections. The third (“Zhuang 3”) has 23 sections. The last, which the reviewer is commenting on (“Zhuang 4”), only has 3 sections. The amount of brain area covered by these three sections is significantly limited compared with the other animals. Therefore, the amount of cluster overlap with the other three animals is necessarily much smaller. Another salient fact is that Zhuang 1 and 2 are collected coronally, and Zhuang 3 and 4 are collected sagittally. The sections corresponding to Zhuang 4 (the animal with the fewest sections collected) only correspond to one hemisphere of the brain and only the most lateral side. Similarly, Zhuang 4 only has 162,579 cells. The other animals each contain more than 1M cells (2.8M for Zhuang 1, 1.2M for Zhuang 2, and 1.5M for Zhuang 3). We believe it is reasonable not to observe complete cluster agreement in this context. To make this clearer, we have emphasized this fact in both the figure caption and the Results section.

While the reviewer brings up an important point, the focus of our study is on identifying coherent clusters across animals. We note that it is not necessarily a given that fitting clusters on concatenated animals will produce integrated clustering. Additionally, we note that in Figure 6c, we fit domains on each animal's dataset separately and then compared to CCF, finding that even when domains were fit only on one animal of Zhuang 1, 2, and 3, correspondence with CCF was high. We also note that our approach of benchmarking integration by clustering across samples, specifically when samples are all included in the same clustering run, is based on a number of works in the field works such as Luecken et al. (Nature Methods, 2022)[1]. This perspective is also shared by a number of other methods specifically for integration, not limited to [2], [3], and [4].

Regarding the choice to use $k=1300$ throughout the manuscript, we point to two important details. First, we emphasize through our analyses in Figures 3, 4, and 5 that spatial clusters identified at $k=1300$ concord strongly with known literature results in subiculum and multiple subregions of superior colliculus. Secondly, we emphasize our stability and goodness-of-fit analyses in the previous draft **Supplementary Figure 8**, which support our choice to analyze the Allen 1 dataset at a higher resolution.

[1] Luecken, M. D., Buttner, M., Chaichoompu, K., Danese, A., Interlandi, M., Mueller, M. F., Strobl, D. C., Zappia, L., Colome-Tatche, M., Theis, Fabian. Benchmarking atlas-level data integration in single-cell genomics. Nature Methods, 2023.

[2] Hie, B., Bryson, B. Berger, B. Efficient integration of heterogenous single-cell transcriptomes using Scanorama. Nature Biotechnology, 2019.

[3] Korsunsky, I., Millard, N., Fan, J., Slowikowski, K., Zhang F., Wei, K., Baglaenko, Y., Brenner, M., Loh, P., Raychadhuri, S. Fast, sensitive and accurate integration of single-cell data with Harmony. Nature Methods, 2019.

[4] Cui, H., Wang, C., Maan, H., Pang, KK., Luo, F., Duan, N., Wang, B. scGPT: toward building a foundation model for single-cell multi-omics using generative AI. Nature Methods, 2024.

Minor

-The term 'cell types' is applied to the cluster level data ('5274 cell types' in Figure 2g for example).

We thank the reviewer for their comment and realize that the reference to "clusters" on line 228 may lead to some confusion. The "clusters" we refer to here are the name for the finest level of cell type annotations in the ABC-MWB (for more context, the coarsest level of cell type annotations are called "classes", the next finer level refers to "subclasses", etc.). We have attempted to clarify this in the text as we agree that the phrasing in the original text may lead to unnecessary confusion:

"We also investigated correspondence of cell type composition with more granular single cell annotations, employing the lowest-level single cell annotations (the "cluster" level, with 5274 cell types, as opposed to the "subclass" level with 338) from ABC-MWB."

-No classifier loss curve is provided. Does the loss plateau, or is more data needed to predict the expression of the focus cell?

We thank the reviewer for this comment and agree that this information would be helpful to include in the paper. To address this, we have included the following supplementary figure which shows that validation loss is saturating for our Allen 1 dataset training run.

Supplementary Figure 1. Train and validation loss curves for CellTransformer trained on the Allen 1 dataset.

Reviewer #3 (Remarks to the Author):
